# CD301b[+] dendritic cells suppress T follicular helper cells and antibody responses to protein antigens

Yosuke Kumamoto[1,2], Toshiro Hirai[3,4], Patrick W Wong[1,2], Daniel H Kaplan[3,4], Akiko Iwasaki[1,2]*

[1]Department of Immunobiology, Yale University School of Medicine, New Haven, United States; [2]Howard Hughes Medical Institute, Yale University School of Medicine, New Haven, United States; [3]Department of Dermatology, University of Pittsburgh, Pittsburgh, United States; [4]Department of Immunology, University of Pittsburgh, Pittsburgh, United States

**Abstract** Strong antibody response is considered a hallmark of a successful vaccine. While dendritic cells (DCs) are important for T follicular helper (Tfh) cell priming, how this process is regulated in vivo is unclear. We show here that the depletion of CD301b[+] DCs specifically enhanced the development of Tfh cells, germinal center B cells and antibody responses against protein antigens. Exaggerated antibody responses in mice depleted of CD301b[+] DCs occurred in the absence of any adjuvants, and resulting antibodies had broader specificity and higher affinity to the immunogen. CD301b[+] DCs express high levels of PD-1 ligands, PD-L1 and PD-L2. Blocking PD-1 or PD-L1 during priming in wild-type mice partially mimicked the phenotype of CD301b[+] DC-depleted animals, suggesting their role in Tfh suppression. Transient depletion of CD301b[+] DC results in the generation of autoreactive IgG responses. These results revealed a novel regulatory mechanism and a key role of CD301b[+] DCs in blocking autoantibody generation.

*For correspondence: akiko.iwasaki@yale.edu

**Competing interests:** The authors declare that no competing interests exist.

## Introduction

Protective adaptive immunity relies on cellular and humoral immune responses. Through presenting processed antigens on their MHC molecules, dendritic cells (DCs) are primarily responsible for initiating T cell-mediated cellular immunity, which is also required for T cell-dependent antibody responses. Germinal centers (GCs) are specialized microarchitecture within the follicles in the secondary lymphoid organs, where B cells undergo affinity maturation, selection and proliferation. Follicular helper T cells (Tfh) are a subset of CD4[+] T cells marked by their expression of inhibitory receptor PD-1 and chemokine receptor CXCR5 and play important roles in the development of GC B cells by providing differentiation, survival and proliferation signals (*Crotty, 2011*; *Ma et al., 2012*). While co-maturation between GC B cells and Tfh is required to stably maintain both subsets (*Baumjohann et al., 2013*; *Deenick et al., 2010*; *Qi et al., 2008*), it has been shown that early instructive signals from antigen-bearing DCs are both necessary and sufficient to prime Tfh differentiation (*Baumjohann et al., 2011*; *Choi et al., 2011*; *Deenick et al., 2010*; *Goenka et al., 2011*).

For the development of antibody responses, DCs not only promote Tfh differentiation but also directly support B cell survival and class-switching through their expression of B cell stimulatory ligands such as BAFF and APRIL (*MacLennan and Vinuesa, 2002*). Moreover, DCs are also important as antigen carriers that provide the initial antigen-specific receptor signaling to the cognate B cells in the lymph nodes (LNs) (*Qi et al., 2006*). Indeed, in vivo antigen delivery to receptors specifically expressed by DCs can trigger robust class-switched antibody production even without an

**eLife digest** The immune system consists of cells and biological processes that defend the body against infection and disease. Molecules called antibodies are a key component of most immune responses because they can bind to invading microbes and either neutralize them or mark them for destruction by immune cells. In autoimmune diseases, antibodies may also target the body's own cells, with the result that the immune system mistakenly attacks tissues and organs.

Immune cells called T follicular helper cells activate the cells that produce antibodies, the B cells. The T follicular helper cells are themselves activated by other cells called dendritic cells, but there are many types of dendritic cells that activate a range of different immune cells. It was not clear whether there are types of dendritic cell that specifically activate – or suppress – the responses of T follicular helper cells.

To investigate this issue, Kumamoto et al. studied both normal mice and mice in which the particular types of dendritic cell can be depleted. The investigation revealed that dendritic cells that produce a protein called CD301b suppress the activity of T follicular helper cells. In mice with few CD301b-producing dendritic cells, the activity of the T follicular helper cells was higher than in normal mice, resulting in the hyperactivation of the B cells. This also led to the antibodies behaving in exaggerated ways, as can occur in autoimmune diseases. Other experiments that depleted some other types of dendritic cell – but not those that produce CD301b – completely abolished the antibody response of the mice.

Future experiments are now needed to investigate exactly how the CD301b-producing dendritic cells block the activity of T follicular helper cells. Exploiting this mechanism could help to design treatments for autoimmune diseases.

adjuvant (*Chappell et al., 2014*; *Xu and Banchereau, 2014*). Importantly, antigen delivery strategies targeting distinct DC subsets seem to result in increased antigen-specific antibody titers regardless of the DC subset targeted (*Lahoud et al., 2011*; *Yao et al., 2015*), suggesting that multiple DC subsets are capable of inducing antibody production (*Chappell et al., 2014*). However, the regulatory mechanism that counteracts the ability of DCs to induce humoral responses is less well understood.

In mouse skin, there are at least three distinct skin-resident DC subsets at the steady-state including epidermal Langerhans cells (LCs), Langerin$^+$ CD103$^+$ dermal DCs (DDCs) and Langerin$^-$ CD11b$^+$ DDCs (*Kaplan et al., 2012*). CD301b$^+$ DCs are a major subset of migratory CD11b$^+$ DDCs that express the highest levels of PD-1 ligands, PD-L1 and PD-L2, among DC subsets in the skin-draining LNs (dLNs) (*Gao et al., 2013*; *Kumamoto et al., 2009*, *2013*; *Murakami et al., 2013*). In addition to the dermis, CD301b$^+$ DCs are found in the submucosa of gastrointestinal, respiratory and reproductive tract (*Denda-Nagai et al., 2010*; *Kumamoto et al., 2013*). By using Mgl2-DTR mice in which one of the endogenous *Mgl2* alleles (encoding CD301b protein) are replaced with a diphtheria toxin receptor (DTR)-GFP cassette, we have shown previously that inducible depletion of CD301b$^+$ DCs by injecting diphtheria toxin (DT) results in impaired Th2 differentiation of antigen-specific CD4$^+$ T cells following footpad immunization of protein antigen with Th2 adjuvants as well as after subcutaneous infection with *Nippostrongylus brasiliensis*. Notably, despite the impaired Th2 differentiation, mice that lack CD301b$^+$ DCs developed intact antibody responses against *N. brasiliensis* infection (*Kumamoto et al., 2013*). In addition, recent studies have expanded the role of CD301b$^+$ DCs beyond the Th2 differentiation program, by demonstrating that they are required for IL-17 production from dermal γδT cells following epidermal infection with *Candida albicans,* or from Th17 cells with intranasal infection with *Streptococcus pyogenes* (*Kashem et al., 2015b*; *Linehan et al., 2015*).

Here, we describe the role of CD301b$^+$ DC in the regulation of humoral immunity. We show that CD301b$^+$ DC depletion results in a marked increase in Tfh, GC B cell and antibody responses to protein antigens even in the absence of adjuvants. Acute antibody blockade of PD-L1, but not PD-L2, at the time of vaccination enhanced Tfh, GC B and antibody responses in CD301b$^+$ DC-dependent manner. In addition, transient depletion of CD301b$^+$ DCs resulted in the generation of autoreactive antibody responses. Our study reveals a role for CD301b$^+$ DCs in negative control of humoral responses, and has important implications in vaccine design and autoimmunity.

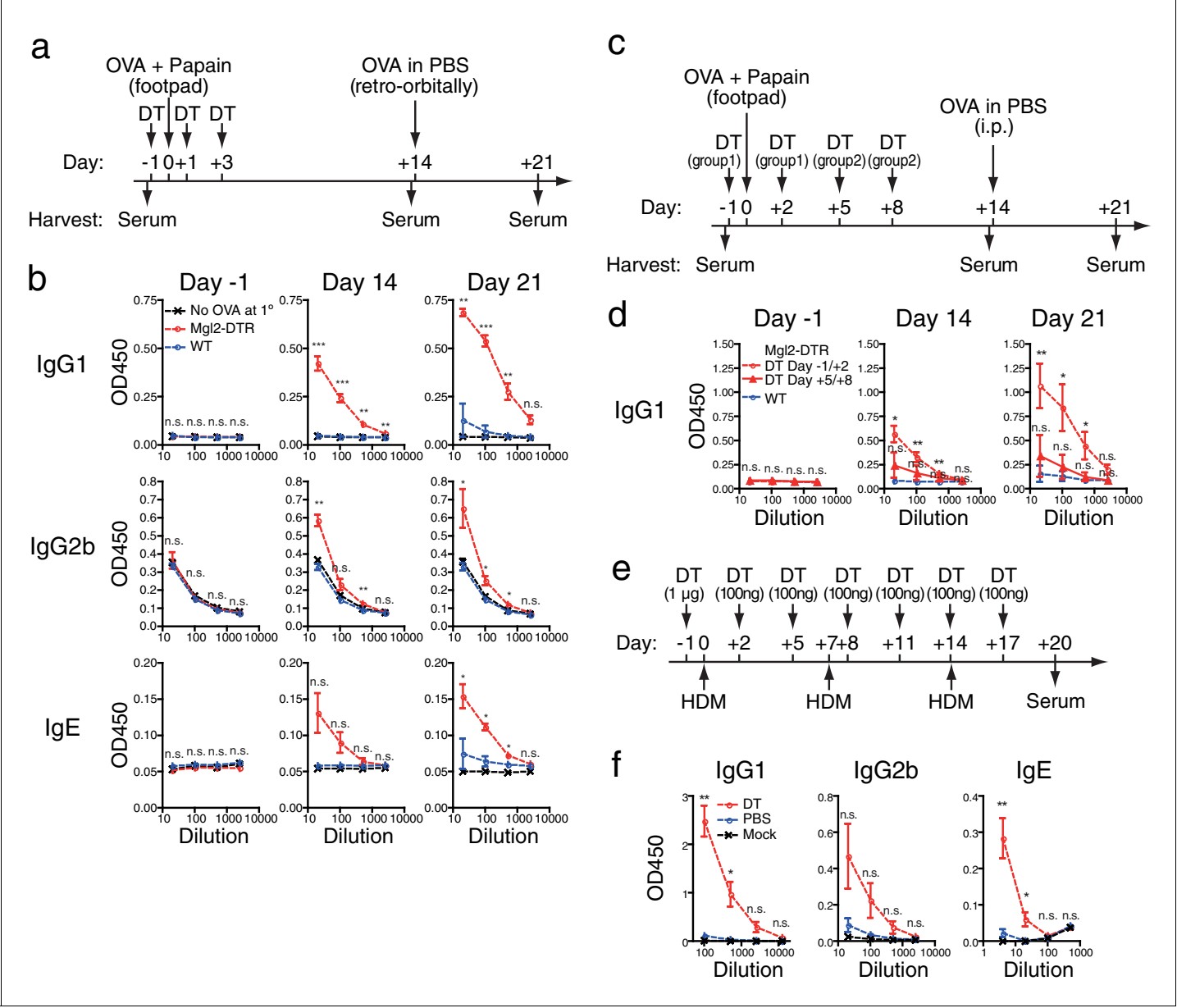

**Figure 1.** Depletion of CD301b⁺ DCs leads to enhanced antibody responses. (a,b) WT and Mgl2-DTR mice were injected i.p. with 0.5 μg DT on days −1, +1 and +3. Mice were immunized with 50 μg papain in 20 μl phosphate-buffered saline (PBS) with or without (No OVA at 1°) 5 μg OVA in the footpad. On day 14, OVA (50 μg in 100 μl PBS) was injected retro-orbitally without adjuvant. Sera were harvested on days −1, +14 and +21. OVA-specific antibody titers were detected by ELISA. Bars indicate mean ± S.E.M. calculated from three (WT and Mgl2-DTR) or two (No OVA at 1°) individual mice. Representative data from three independent experiments are shown. (c,d) Mgl2-DTR mice were treated with DT and immunized with 5 μg OVA and 50 μg papain in the footpad as in a (DT Day −1/+2). Alternatively, WT or Mgl2-DTR mice were immunized with OVA and papain on day 0, then treated with DT on days five and eight (DT Day +5/+8). All received a boost immunization with OVA in PBS on day 14. Sera were collected in two independent experiments from total of 5–6 mice per group. (e,f) Mgl2-DTR mice were treated i.p. with DT or PBS and painted with whole cell lysates of house dust mite (HDM) for three times as shown in e. HDM-specific antibody titers were tested by ELISA against the whole cell HDM lysates (f). Data were collected from five animals in each group and are representative of two independent experiments. Bars indicate mean ± S.E.M. n.s., not significant, *p<0.05, **p<0.01, ***p<0.001 by two-tailed Student's t-test. All statistics indicate comparison to the WT or undepleted control shown in each panel.

## Results

### Depletion of CD301b+DCs enhances antigen-specific class-switched antibody production in response to type 2 immunogens

To understand the role of CD301b[+] DC on the antibody response, we utilized a model of a single immunization with OVA and papain in the footpad, which by itself induces a minimal antibody response in wild-type (WT) mice (*Figure 1a*). Immunization of Mgl2-DTR mice depleted of CD301b[+] DCs resulted in greatly enhanced production of OVA-specific class-switched antibodies (*Figure 1b*). The increased antibody titers were evident on day 14 after a single immunization and enhanced following a systemic secondary exposure to the same antigen without an adjuvant (*Figure 1b*). However, the antibody titers were not elevated when CD301b[+] DCs were depleted five days post-immunization (*Figure 1c,d*). These results indicated that the presence of CD301b[+] DCs during the early phase of primary immunization has a negative and lasting impact on humoral immunity.

To test the role of CD301b[+] DCs in antibody production against another allergen, we sensitized CD301b[+] DC-depleted mice epicutaneously by painting whole extracts from house dust mite *Dermatophagoides pteronyssinus* (*Figure 1e*). Although this needle-free immunization protocol induced minimal amount of antibodies in WT mice even after multiple exposure to the antigen, CD301b[+] DC-depleted mice generated robust antibody production (*Figure 1f*). These results indicate that the enhanced antibody responses in CD301b[+] DC-depleted mice are found in response to subcutaneous papain immunization and to epicutaneous allergen exposure.

### Depletion of CD301b+DCs enhances antigen-specific antibody production in response to intraperitoneal immunization

To examine if antibody responses primed in organs other than the skin were also affected by the depletion of CD301b[+] DCs, we next immunized Mgl2-DTR mice intraperitoneally (i.p) with OVA and alum, another type 2 adjuvant that requires CD301b[+] DCs for Th2 differentiation (*Kumamoto et al., 2013*). We observed a trend for enhanced antibody production in CD301b[+] DC-depleted animals, which was likely a result of the depletion of CD301b[+] DC in the peritoneal cavity and its draining mediastinal LN (*Kool et al., 2008*) (*Figure 2*). As in the skin-dLNs, CD301b[+] DCs in the peripheral organs expressed CD11b and MHCII but not Ly6C, but were nearly absent from the spleen (*Figure 2c–f*). To further characterize the CD301b[+] DCs in the skin-dLNs and the peritoneum, we examined the expression of other molecules in CD301b[+] cells in these organs (*Figure 2—figure supplement 1*). In the skin-dLNs, the majority of CD301b[+] cells were consistent with migratory DCs, as they were MHCII[hi]CD11c[+], CD64[lo]Ly6C[lo], CD11b[+]CD11c[+], CD24[lo]CD11b[+], and expressed a conventional DC marker Zbtb46 (*Meredith et al., 2012*; *Satpathy et al., 2012*). CD301b[+] cells in the peritoneal cavity had a similar phenotype, though their Zbtb46 expression levels were lower. The DT treatment of Mgl2-DTR mice resulted in specific deletion of the CD301b[+] DC subset, and did not grossly affect other cell types, except for an increase in the proportion of Ly6C[+] cells in the peritoneal cavity (*Figure 2—figure supplement 1*). These results indicate that CD301b[+] DCs constitute an important migratory DC population in skin and peritoneal cavity that negatively regulate antibody responses.

### CD301b+ DC depletion mitigates the requirement for adjuvant for antibody responses

Next, we examined the parameters controlled by CD301b[+] DCs. One possibility is that CD301b[+] DCs regulate antibody responses by limiting the amount of antigens available for initiating a humoral response. If this were the case, increasing the antigen dose should overcome the suppression by these DCs. To address this possibility, we immunized DT-treated Mgl2-DTR mice with different doses of OVA with a fixed amount of papain (*Figure 3*). To account for any potential effects of the DT treatment, we treated Mgl2-DTR mice with a nontoxic DT mutant, CRM197 (*Malito et al., 2012*), and used this group as the undepleted control. Increasing doses of the antigen did not overcome suppression of antibody responses by CD301b[+] DCs (*Figure 3b*), indicating that limiting antigen access is not the dominant mode of antibody suppression by these DCs.

Next, we examined the requirement for adjuvants. Notably, CD301b[+] DC-depleted mice developed strong antibody responses against OVA even when they were immunized without any

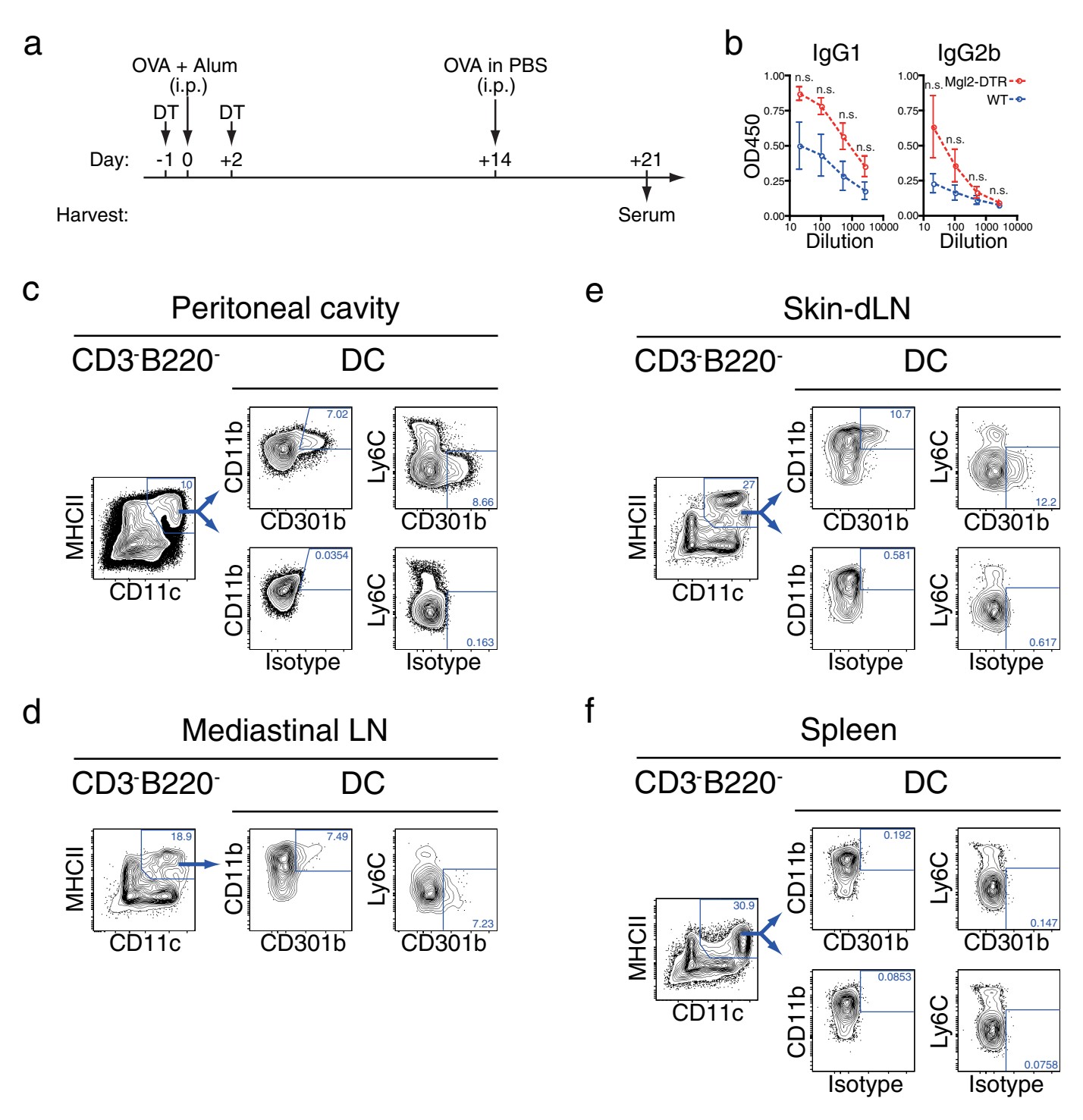

**Figure 2.** Depletion of CD301b[+] DCs enhances antibody responses in the non-skin dLNs. (a,b) Mice were immunized with 10 µg OVA in 100 µl alum i.p. and received i.p. injection of 10 µg OVA without adjuvant on day 14. Sera were harvested on day 21 for OVA-specific antibody ELISA. Bars indicate mean ± S.E.M. calculated from three animals. (c–f) Indicated organs and peritoneal exudate cells were harvested from naïve WT mice and analyzed for surface CD301b expression along with indicated markers. 'Isotype' indicates binding of non-specific rat IgG2a conjugated with the same fluorochrome as the anti-CD301b mAb used.

The following figure supplement is available for figure 2:

**Figure supplement 1.** Phenotype of CD301b[+] DCs in LNs and peritoneal cavity.

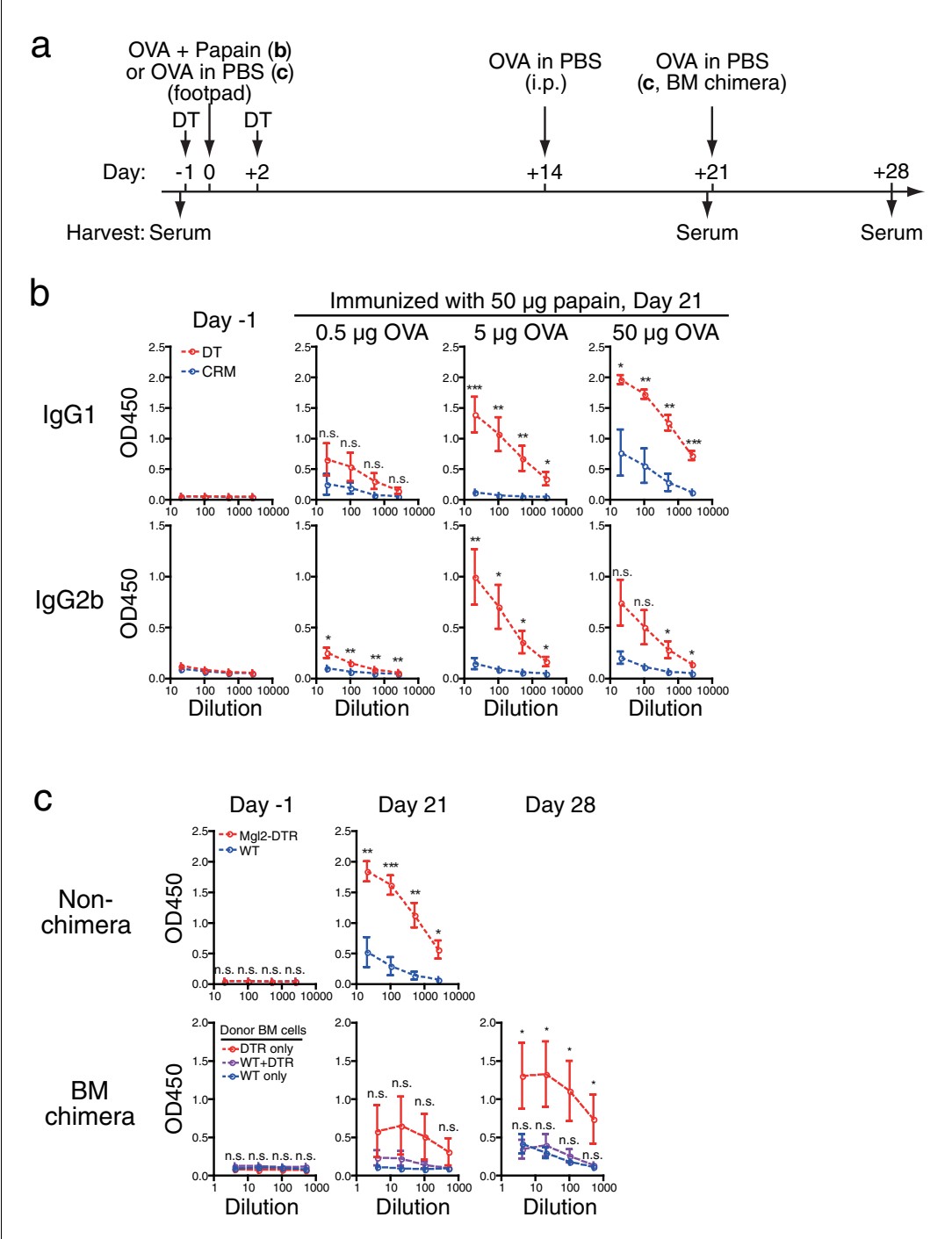

**Figure 3.** Depletion of CD301b[+] DCs leads to enhanced antibody responses to the antigen immunized with weak or no adjuvant. (a) Immunization and sample collection timeline. (b) Mgl2-DTR mice were treated with 0.5 µg DT or its inactive mutant CRM197 (CRM) on days -1 and +2 and immunized on day 0 with 50 µg papain plus indicated amount of OVA in the footpad. All mice received i.p. injection of 10 µg OVA without adjuvant on day 14 and sera were harvested on day 21 for OVA-specific antibody ELISA. (c) WT and Mgl2-DTR mice were treated with 0.5 µg DT on days −1 and +2 and immunized in the footpad with 5 µg OVA without any adjuvant on day 0 (top). Alternatively, lethally-irradiated WT mice were reconstituted with Mgl2-DTR (DTR only), WT (WT only) or 1:1-mixture of WT and Mgl2-DTR (WT+DTR) BM cells, then immunized with 5 µg OVA without adjuvant (bottom). All mice received i.p. injection of 10 µg OVA without adjuvant on day 14 and day 21, and sera were harvested on day 21 and day 28 for OVA-specific IgG1 ELISA. Bars indicate mean ± S.E.M. calculated from 4–11 individual mice. n.s., not significant, *p<0.05, **p<0.01, ***p<0.001, ****p<0.0001 by two-tailed Student's t-test. All statistics indicate comparison to the undepleted (CRM-treated, WT or WT only BM chimera) control shown in each panel.

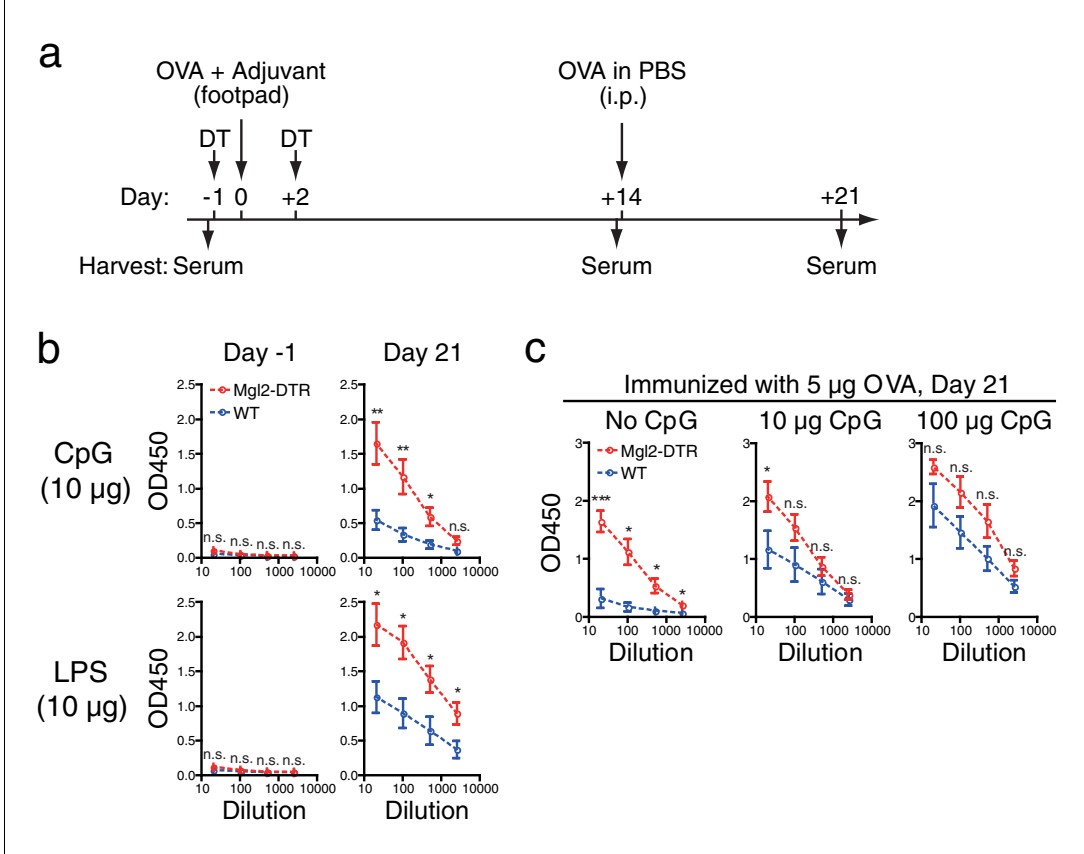

**Figure 4.** Depletion of CD301b+ DCs enhances antibody responses to type 1 immunization when adjuvant is limited. WT and Mgl2-DTR mice were treated with 0.5 μg DT on days −1 and +2 and immunized in the footpad with 5 μg OVA plus indicated amount of CpG2216 or LPS as in **a**. All mice received i.p. injection of 10 μg OVA without adjuvant on day 14 and sera were harvested on day 21 for OVA-specific IgG2b ELISA. Sera were collected in two to four independent experiments. Bars indicate mean ± S.E.M. calculated from 6–7 (**b**) or 4–11 (**c**) individual mice. n.s., not significant, *p<0.05, **p<0.01 by two-tailed Student's t-test. All statistics indicate comparison to the WT control shown in each panel.

adjuvants (*Figure 3c*). A possible explanation of these results is that the CD301b+ DCs killed by DT release endogenous materials capable of serving as an adjuvant to stimulate antibody responses to OVA. To test this possibility, we next immunized lethally-irradiated WT mice reconstituted with a 1:1 mixture of WT and Mgl2-DTR bone marrow (BM) cells. In order to avoid any DT-induced adjuvanticity masked by the presence of exogenous adjuvant, we took advantage of the adjuvant-free immunization approach. Although antibody levels in BM chimeric mice were generally lower than in non-irradiated mice, prime (day 0) and 2 boosts (days 14 and 21) with adjuvant-free OVA resulted in significantly increased OVA-specific antibody titers in mice reconstituted only with Mgl2-DTR BM cells (*Figure 3c*). In contrast, mice reconstituted with a mixture of WT and Mgl2-DTR yielded minimal antibody production (*Figure 3c*), indicating that the remaining WT CD301b+ DCs were sufficient to suppress humoral immunity in such BM chimeric mice even if the dying CD301b+ DCs released endogenous adjuvants. Collectively, these results indicate that materials released from dying CD301b+ DC cannot account for the enhanced antibody responses, and that lifting suppression by CD301b+ DCs enables induction of antibody to soluble protein antigens in the absence of adjuvants.

## Type 1 adjuvants overcome suppressive effects of CD301b+ DCs for humoral immunity

Thus far, our results indicate that humoral immunity induced by type 2 adjuvant or in the absence of adjuvant is suppressed by CD301b+ DCs. To address if CD301b+ DCs exert same suppressive effects in the presence of Th1-inducing adjuvants, we next examined antibody responses in CD301b+ DC-

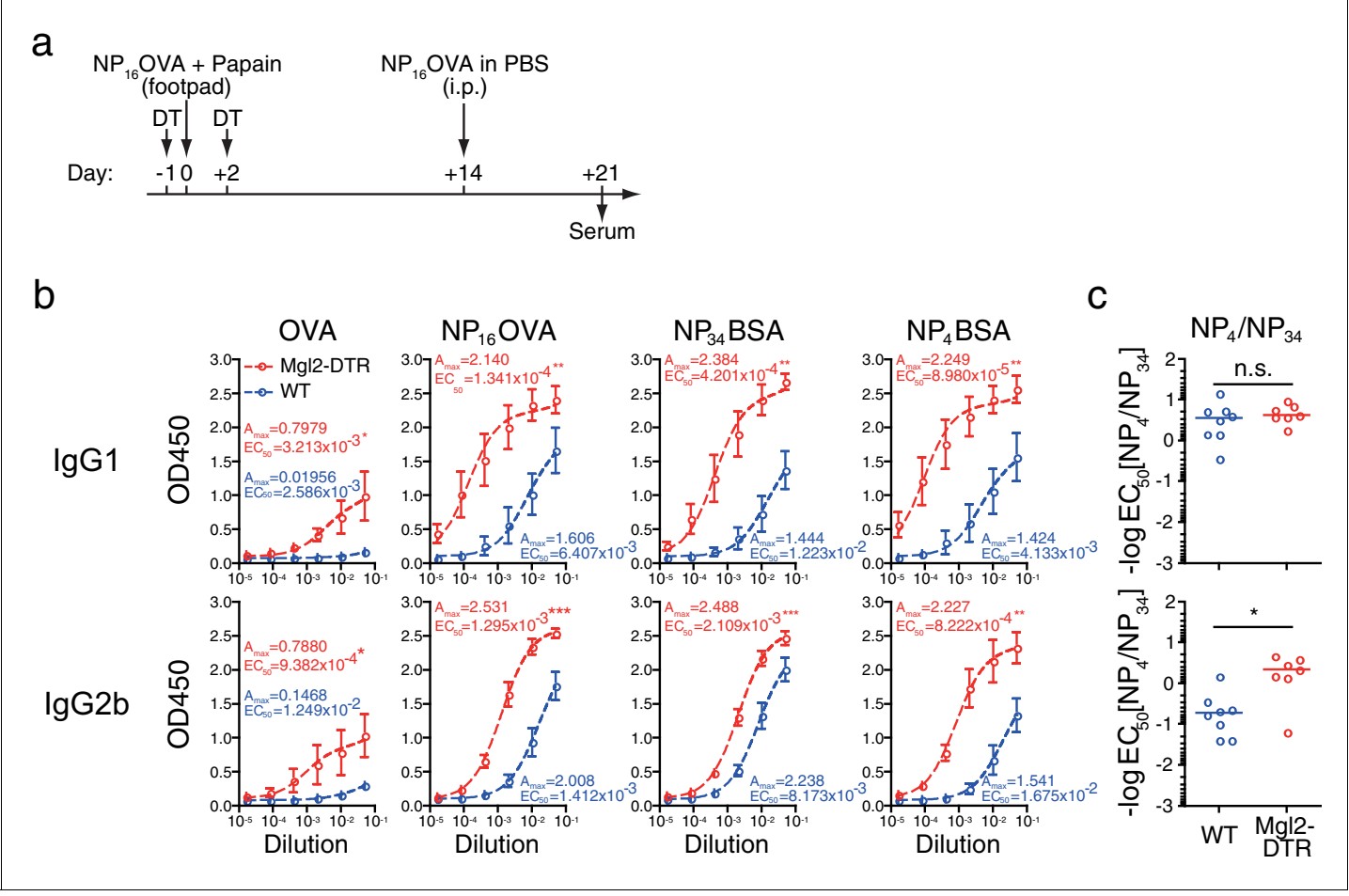

**Figure 5.** Depletion of CD301b+ DCs enhances the development of antigen-specific B cells with GC B cell phenotype without negatively affecting the affinity of antibodies. As in **a**, mice were treated with DT and immunized with 5 μg NP$_{16}$OVA plus 50 μg papain in the footpad. Two weeks later, all mice received i.p. injection of NP$_{16}$OVA in PBS. (**b**) Serum antibodies against indicated antigens were measured by ELISA. n = 7–8 per group collected in two independent experiments. A$_{max}$ and EC$_{50}$ are shown in units of OD450 and serum dilution, respectively. (**c**) The ratio between EC$_{50}$ for the binding to NP$_4$ epitope and that for NP$_{34}$ epitope was calculated in each mouse. n.s., not significant, *p<0.05, **p<0.01, ***p<0.001 by Mann-Whitney test. Bars indicate means ± S.E.M. (**b**) or medians (**c**).

depleted mice after immunization with two kinds of type 1 adjuvant, CpG and LPS. CD301b+ DC-depleted animals produced higher levels of OVA-specific IgG2b, suggesting that CD301b+ DCs also play a regulatory role in antibody production against type 1 antigens (*Figure 4a,b*). However, increasing the dose of CpG from 10 μg to 100 μg abrogated the difference in antibody levels between WT and CD301b+ DC-depleted mice (*Figure 4c*). These results indicate that the suppressive function of CD301b+ DCs can be overcome by the use of high dose type 1 adjuvants.

## Depletion of CD301b+ DCs broadens antibody specificity and enhances affinity

To address the potential impact of the depletion of CD301b+ DCs on the specificity and affinity of the antibodies, we next immunized the mice with papain plus OVA conjugated with 4-hydroxyl-3-nitrophenyl acetyl in 1:16 ratio (NP$_{16}$OVA), followed by a secondary immunization with NP$_{16}$OVA without adjuvant through a systemic route (*Figure 5a*). WT mice immunized with this protocol developed a significant amount of NP$_{16}$OVA-specific antibodies, albeit to a lesser extent than the CD301b+ DC-depleted mice. However, consistent with previous studies showing preferential development of hapten-specific rather than carrier-specific antibodies upon immunization with

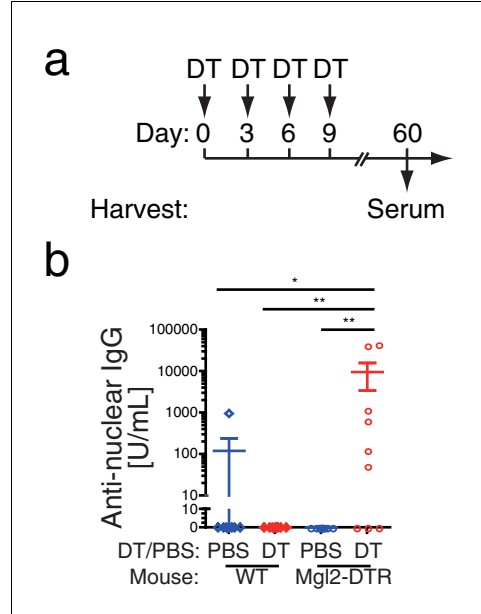

**Figure 6.** Depletion of CD301b$^+$ DCs enhances autoreactive antibodies. WT or Mgl2-DTR mice were treated i.p. with PBS or 0.5 μg DT every third day for 9 days as in **a**. Sera were harvested at day 60 and examined for anti-nuclear IgG antibodies by ELISA (**b**). Bars indicate mean ± S.E.M. *p<0.05, **p<0.01 by Mann-Whitney test.

haptenated proteins (*Palm and Medzhitov, 2009*; *Shimizu et al., 2007*), the majority of the NP$_{16}$OVA-specific antibodies produced in WT mice were directed to the hapten, as anti-OVA antibody responses were undetectable (*Figure 5b*). In contrast, CD301b$^+$ DC-depleted mice developed significantly higher amount of antibodies against all epitopes examined. Notably, unlike WT animals, CD301b$^+$ DC-depleted mice also produced antibodies specific to the OVA carrier, indicating that the specificity of the antibodies were broadened in these mice (*Figure 5b*). In addition, the ratio between the binding of the antibodies to bovine serum albumin (BSA) conjugated with high (NP$_{34}$BSA) and low (NP$_4$BSA) numbers of NP suggested that antibodies in immunized CD301b$^+$ DC-depleted mice underwent similar (IgG1) or even slightly higher (IgG2b) levels of affinity maturation compared to the WT mice (*Figure 5c*). These results indicate that CD301b$^+$ DCs suppress the magnitude, breadth and affinity of antibodies.

## Depletion of CD301b$^+$ DCs enhances autoreactive antibodies

Based on the results above, we questioned why such antibody-suppressive DCs exist in mice. One possibility is that the CD301b$^+$ DCs are needed to suppress the generation of autoreactive antibodies. Indeed, transient depletion of CD301b$^+$-DCs for 9 days resulted in the development of anti-nuclear IgG antibodies 60 days later (*Figure 6*). These results indicate that CD301b$^+$ DCs play a key role in suppressing the emergence of autoantibodies.

## CD301b$^+$ DCs regulate expansion of antigen-specific germinal center B cells

To understand the cellular basis for the enhanced antibody production generated in the absence of CD301b$^+$ DCs, we next analyzed the cellular composition in the popliteal dLN (right) and non-draining (nd) (left) popliteal LN during the priming phase after a single immunization with OVA and papain in the right footpad (*Figure 7a*). Consistent with our previous report, the size of the dLNs at three days post-immunization in CD301b$^+$ DC-depleted animals was much smaller than in undepleted controls, likely due to impaired screening and retention of naïve CD4$^+$ T cells (*Kumamoto et al., 2013*) (*Figure 7—figure supplement 1a,b*). However, the number of B cells in the dLNs started to increase on day five in the CD301b$^+$ DC-depleted animals compared to the WT mice, which continued to increase until day seven (*Figure 7—figure supplement 1c*). Consistent with the higher class-switched high-affinity antibody titers (*Figures 1–3*, *5*), the increase in the number of GC B cells (CD95$^+$ GL7$^+$) was even more exaggerated than the increase in the number of total B cells in CD301b$^+$ DC-deleted mice (*Figure 7b* and *Figure 7—figure supplement 1e*).

We next examined the phenotype of B cell subsets in the dLNs 14 days after a single immunization with OVA and papain in the footpad (*Figure 7c*). In CD301b$^+$ DC-depleted animals, an increased fraction of B cells that downregulated surface IgD receptor and bound to soluble fluorescently-labeled OVA protein was found, indicative of OVA-specific activated B cells (*Figure 7d,f*). Some IgD$^-$ cells expressed a plasma cell marker CD138 (*Figure 7e,g*). We also observed an expansion of the CD38$^-$GL7$^+$ germinal center (GC) B cell fraction, while the CD38$^+$GL7$^-$ naïve/memory fraction (*Taylor et al., 2015*) was significantly reduced (*Figure 7h–j*). Notably, OVA-specific B cells were

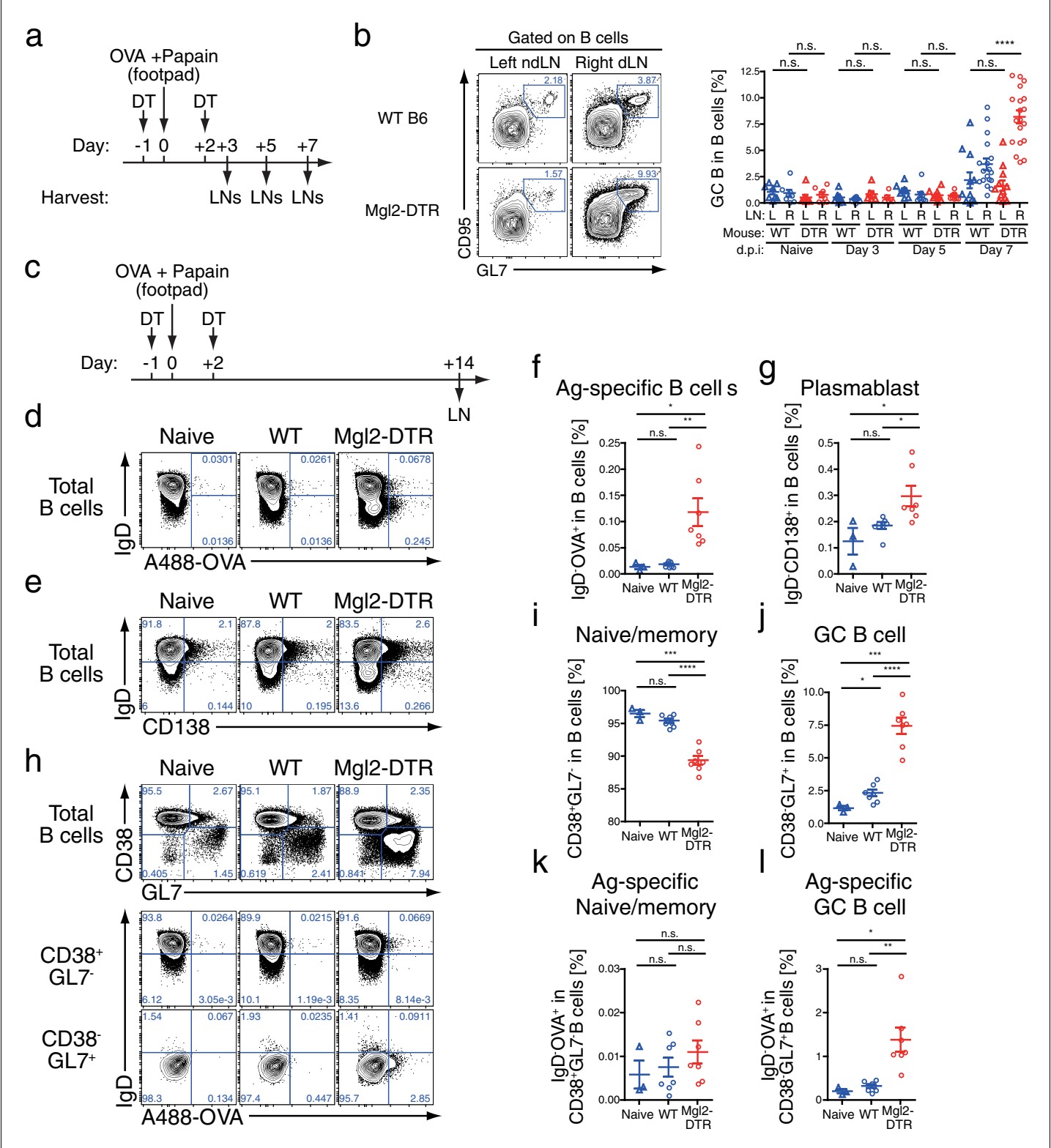

**Figure 7.** Depletion of CD301b[+] DCs leads to GC B cell expansion in response to immunization with OVA and papain. (**a,b**) WT and Mgl2-DTR (DTR) mice were treated with DT on days −1 and +2 and immunized with 50 µg papain and 5 µg OVA in 20 µl PBS in the right footpad. The left non-draining (L) or right draining (R) popliteal LNs were harvested on indicated days post immunization (d.p.i) and analyzed for the development of GC B cells (GL7[+] CD95[+]). Mice that were treated with DT but left unimmunzed were analyzed at 7 d.p.i. (eight days from the first DT treatment) and are indicated as 'Naïve'. Flow-cytometry panels in **b** are representative of data from day seven. (**c**) WT and Mgl2-DTR mice were treated with DT on days −1 and +2 and

*Figure 7 continued on next page*

Figure 7 continued

immunized on day 0 with 5 µg OVA with 50 µg papain as in *Figure 2*. On day 14, dLNs were harvested and cells were stained for the cell-surface markers as well as for their binding to Alexa488-conjugated OVA protein (A488-OVA). (d–g) Intracellular binding to A488-OVA (d,f) and CD138 expression (e,g) by activated IgD⁻ B cells. (h–j) CD38 and GL7 expression in total B cells. (k,l) Cell-surface IgD expression and intracellular A488-OVA binding in naïve/memory (CD38⁺GL7⁻) B cells (k) and GC (CD38⁻GL7⁺) B cells (l) gated as in e. In all graphs, each dot indicates an individual mouse and the data were pooled from two independent experiments. n.s., not significant, *p<0.05, **p<0.01, ***p<0.001, ****p<0.0001 by two-tailed Student's t-test. Bars indicate means ± S.E.M.

The following figure supplement is available for figure 7:

**Figure supplement 1.** Cellular kinetics in CD301b⁺ DC-depleted mice immunized with papain and OVA.

preferentially enriched in the CD38⁻GL7⁺ GC B cell compartment in CD301b⁺ DC-depleted mice (*Figure 7h,k,l*). These results indicate that the depletion of CD301b⁺ DCs induced antigen-specific B cells with a GC B cell phenotype in the dLNs.

## CD301b⁺ DCs control expansion of antigen-specific Tfh cells

The development of GC B cells requires help from Tfh cells through cognate interaction (*Crotty, 2011*; *Ma et al., 2012*). Indeed, CD4⁺ T cells of the Tfh phenotype (CXCR5⁺ PD-1⁺) were also increased in number (*Figure 7—figure supplement 1d*) and frequency (*Figure 8a*) within the dLN of immunized CD301b⁺ DC-depleted mice. Similarly, among CD44^hi effector CD4⁺ T cells, the ratio between PSGL1^lo PD-1⁺ (follicular) cells to PSGL1^hi PD-1⁻ (extrafollicular) cells (*Poholek et al., 2010*) was significantly increased in CD301b⁺ DC-depleted animals compared to WT mice, further confirming the expansion of Tfh cells over other effector CD4⁺ T cells (*Figure 8b*). The accumulation was selective to the canonical Tfh cells that express ICOS and Bcl6 and did not affect the follicular regulatory T (Tfr) cells (*Figure 8c,d*), suggesting that the increased GC B cell development was associated with the Tfh expansion rather than decrease in Tfr population. Interestingly, the increase in the Tfh fraction out of the total CD4⁺ T cells (significantly increased already by day five, *Figure 8a*) preceded that of GC B cell increase (not significantly changed by day seven, *Figure 7b*), suggesting that the depletion of CD301b⁺ DCs induced the Tfh accumulation, which resulted in GC B cell expansion.

## Tfh cells that differentiate in the absence of CD301b⁺ DCs have intact IL-4 and IL-21 secretion

Next, to test whether the depletion of CD301b⁺ DCs induces expansion of antigen-specific Tfh cells, we adoptively transferred OVA-specific OT-II TCR transgenic CD4⁺ T cells into WT or Mgl2-DTR hosts, which were then treated with DT and immunized with OVA and papain. A larger fraction of OT-II CD4⁺ T cells differentiated into Tfh cells in CD301b⁺ DC-depleted hosts than in undepleted hosts (*Figure 8e*). We previously reported OT-II cells primed in the absence of CD301b⁺ DCs failed to differentiate into Th2 cells (*Kumamoto et al., 2013*). Indeed, we observed reduced IL-4 production from CXCR5⁻ OT-II effector T cells in CD301b⁺ DC-depleted mice (*Figure 8f*). How is it possible to have elevated type 2 antibody production in the absence of IL-4-secreting Th2 cells? Although IL-4 is the signature cytokine for Th2 cells, IL-4 is also abundantly produced by Tfh cells and supports production of type 2 antibodies (*King and Mohrs, 2009*; *Reinhardt et al., 2009*). In fact, the expression levels (MFI) of IL-4 and IL-21 in CXCR5⁺ OT-II Tfh cells remained intact in CD301b⁺DC-depleted mice, while the percentages of IL-4- or IL-21-secreting OVA-specific Tfh (CXCR5⁺ OTII) increased (*Figure 8f*). These results indicate that CD301b⁺ DC-depletion results in loss of Th2 cells but increase in Tfh cells capable of secreting IL-4 and IL-21. Further, our data suggest that the expansion of antigen-specific Tfh cells capable of secreting IL-4 and IL-21 underlies the enhanced type 2 antigen-specific antibody and GC B cell responses in the absence of CD301b⁺ DCs.

## CD301b⁺ DCs but not other migratory DCs suppress antibody responses

The results thus far indicated that CD301b⁺ DCs suppress Tfh and GC B cell expansion. Although CD301b-driven DTR-GFP expression is detectable only in a subset of Langerin⁻ CD11b⁺ DCs,

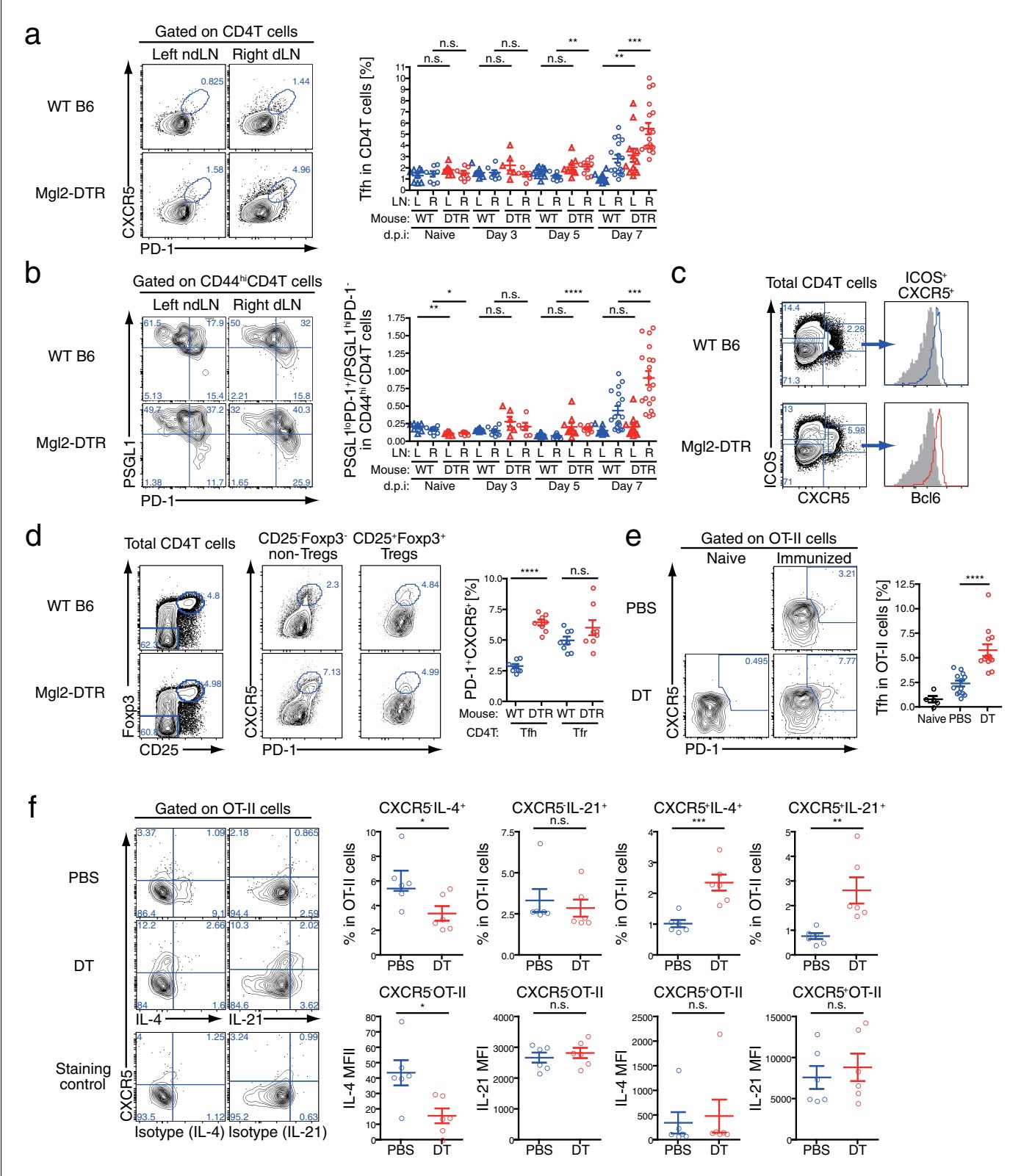

**Figure 8.** GC B cell expansion in CD301b[+] DC-depleted mice is accompanied with Tfh expansion. (**a,b**) Mice were immunized as in *Figure 7a*. Tfh (PD-1[+]CXCR5[+]) differentiation (**a**) and the ratio between follicular (PSGL1[lo]PD-1[+]) and extrafollicular (PSGL1[hi]PD-1[-]) cells within CD44[hi] CD4[+] T cells (**b**) were examined in the draining (R) or non-draining (L) popliteal LNs. Mice that were treated with DT but left unimmunized were analyzed at 7 d.p.i (8 days from the first DT treatment) and are indicated as 'Naïve'. Flow-cytometry panels are representative of data from day seven p.i. (**c,d**) Mice were

*Figure 8 continued on next page*

*Figure 8 continued*

immunized as in **a**. Expression of Bcl6 in ICOS⁺CXCR5⁺CD4⁺ T cells (**c**) and percentages of PD-1⁺CXCR5⁺ cells among Tregs (CD25⁺Foxp3⁺CD4⁺T cells) and non-Tregs (CD25⁺Foxp3⁺CD4⁺T cells) (**d**) in the dLN at 7 d.p.i are shown. Gray histograms in **e** indicate binding of non-specific isotype control mAb binding to ICOS⁺CXCR5⁺CD4⁺ T cells pooled from WT and Mgl2-DTR mice. (**e,f**) Mgl2-DTR mice transferred with CD45.1⁺ OT-II transgenic CD4⁺ T cells were treated with DT or PBS, then immunized as in **a**. On day seven, the draining popliteal LNs were harvested and stained for Tfh markers and cytokines. The donor OT-II cells were identified as CD45.1⁺ CD4⁺ cells. Data were pooled from 2–3 individual experiments per group. Each dot indicates an individual mouse. n.s., not significant, *p<0.05, **p<0.01, ***p<0.001, ****p<0.0001 by two-tailed Student's t-test. Bars indicate means ± S.E.M. MFI, mean fluorescent intensity.

epidermal LCs are also depleted by DT treatment in Mgl2-DTR mice (*Kashem et al., 2015a*; *Kumamoto et al., 2013*). To test whether the enhanced antibody responses in DT-treated Mgl2-DTR mice were due to the depletion of CD301b⁺ DCs or LCs, we used huLangerin-DTR mice, in which DT treatment specifically depletes epidermal LCs without affecting Langerin⁺ CD103⁺ DCs (*Figure 9a,b* and *Figure 9—figure supplement 1a–c*) (*Bobr et al., 2010*). Depletion of LCs did not result in enhanced OVA-specific antibody production (*Figure 9c*). Similarly, depletion of all Langerin⁺ DCs (LCs and Langerin⁺ CD103⁺ DCs) in msLangerin-DTR mice (*Kissenpfennig et al., 2005*; *Poulin et al., 2007*) had no impact on antibody responses (*Figure 9d*). Nevertheless, DCs in general were required for B cell priming, as class-switched antibodies were completely abolished in CD11c-DTR;Mgl2-DTR mice (*Figure 9e*), in which LN-resident DCs were further depleted (*Figure 9—figure supplement 1a–c*). These results indicated that the antibody suppressive property is specific to the CD301b⁺ DCs and is not shared by other migratory DCs.

## LN-resident, but not migratory, DCs promote Tfh, GC B and antibody responses

We next sought to determine which DC subsets are responsible for promoting Tfh and GC B cell development. To first address the role of two other major migratory DCs, we used huLangerin-DTR;Mgl2-DTR and msLangerin-DTR;Mgl2-DTR mice, in which CD301b⁺ DCs and LCs, and Langerin⁺ CD103⁺ DCs in case of the latter, are depleted altogether (*Figure 9b* and *Figure 9—figure supplement 1a–c*). Surprisingly, despite near complete depletion of all migratory DC subsets (*Figure 9—figure supplement 1a,b*), co-depletion of CD301b⁺ DCs, LCs and Langerin⁺ CD103⁺ DCs did not revert the enhanced antibody titers, Tfh, and GC B responses in CD301b⁺ DC-depleted animals (*Figure 9c,d,f*). Similarly, depleting Ly6C⁺ inflammatory monocyte-derived DCs concurrently with the CD301b⁺ DC depletion (*Figure 9—figure supplement 1d*) did not revert the expanded Tfh and GC cells (*Figure 9g*), suggesting that the potential increase in Ly6C⁺ cells in the periphery (*Figure 2—figure supplement 1*) does not directly explain the phenotype. These data are consistent with a model in which CD301b⁺ DCs provide negative signals to Tfh precursors, while non-migrant DC population in the LN promotes Tfh differentiation, GC B and antibody responses.

## CD301b⁺ DCs express PD-1 ligands

PD-1 is an inhibitory receptor that is upregulated in a fraction of activated T cells (*Jin et al., 2011*). PD-1 is also a marker of Tfh cells (*Baumjohann et al., 2011*; *Choi et al., 2011*; *Crotty, 2011*; *Glatman Zaretsky et al., 2009*; *Haynes et al., 2007*). CD301b⁺ DCs express high levels of a PD-1 ligand PD-L2 (B7-DC) (*Gao et al., 2013*; *Kumamoto et al., 2013*; *Murakami et al., 2013*). Indeed, within the non-lymphocyte population in the skin-dLNs, CD301b⁺ DCs express highest levels of both PD-1 ligands PD-L1 (B7-H1) and PD-L2 (*Figure 10a*). More than 90% of all PD-L1⁺ PD-L2⁺ non-lymphocytes were MHCII^hi migratory DCs, of which more than 40% consisted of CD301b⁺ DCs. In addition, a significant number of PD-L1⁺ PD-L2⁺ DCs that do not express CD301b were also present, which was minimally affected by the depletion of CD301b⁺ DCs (*Figure 10—figure supplement 1a*).

## PD-L1 blockade enhances expansion of Tfh and GC B cells only in the presence of CD301b⁺ DCs

To test if PD-1 ligands contribute to the suppression of Tfh and GC B cell expansion by CD301b⁺ DCs, we treated WT or CD301b⁺ DC-depleted mice with anti-PD-L1 or anti-PD-L2 blocking

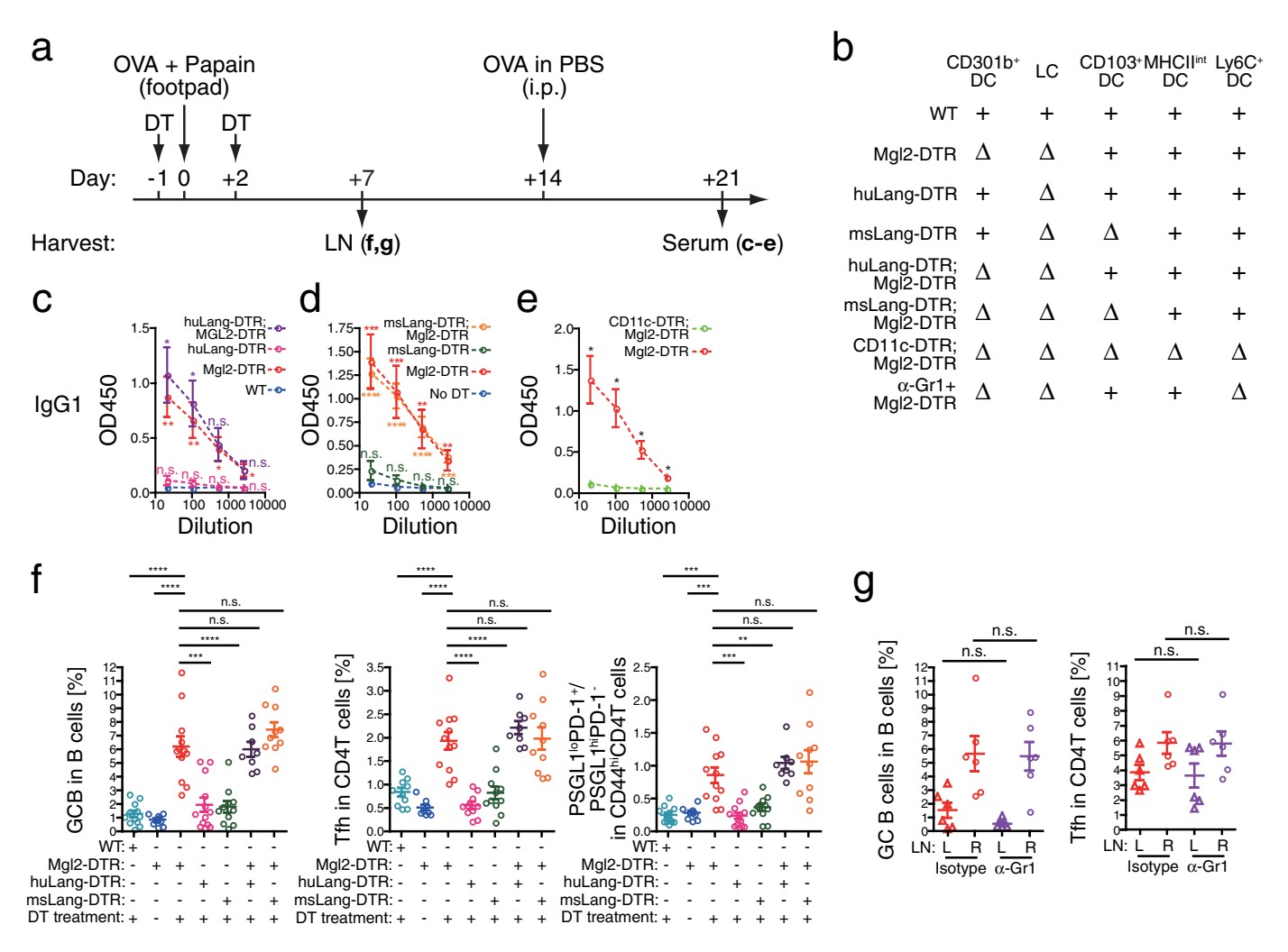

**Figure 9.** Migratory and inflammatory DC subsets are not required for the enhanced antibody, Tfh and GC B cell development induced by the CD301b⁺ DC depletion. (**a–d**) Mice were treated with 0.5 µg DT and immunized with 50 µg papain and 5 µg OVA in the footpad. At the time of immunization, different DC subsets were either intact (+) or depleted (Δ) depending on the host genotype as indicated in **b**. All mice received an i.p. injection of 10 µg OVA in PBS without adjuvant on day 14. Serum OVA-specific IgG1 on day 21 was detected by ELISA. Bars indicate mean ± S.E.M. calculated from 3–6 individual mice in each group. Representative data from 2–3 independent experiments are shown. (**e**) Mgl2-DTR and CD11c-DTR; Mgl2-DTR mice were immunized and boosted as in **a**, except that the mice were treated with a single dose of 125 ng DT on day −1. Sera were harvested on day 21 from three independent experiments and OVA-specific IgG1 was detected by ELISA. Bars indicate mean ± S.E.M. calculated from five (Mgl2-DTR) and three (Mgl2-DTR;CD11c-DTR) individual mice. n.s., not significant, *p<0.05, **p<0.01, ***p<0.001, ****p<0.0001 by two-tailed Student's t-test. All statistics indicate comparison to the undepleted (WT or no DT) control shown in each panel except in **b** and are shown by color-matched asterisks where applicable. (**f**) WT mice and mice that carry Mgl2-DTR, huLang-DTR and/or msLang-DTR in their genetic construct were treated with DT and immunized as in **a**, then LNs were harvested on day seven. Tfh and GC B cell percentages in dLNs. Data were pooled from 2–3 independent experiments per group. (**g**) Mgl2-DTR mice were treated with DT and immunized as in **a**. Mice received i.p. injections of 50 µg anti-Gr-1 (α-Gr1) or Rat IgG2a isotype control antibodies every other day starting on day two. Left non-draining (L) or right draining (R) popliteal LNs were harvested on day seven and analyzed for GC B and Tfh cells. Data were pooled from two individual experiments per group. Each dot indicates an individual mouse. n.s., not significant, *p<0.05, **p<0.01, ***p<0.001, ****p<0.0001 by two-tailed Student's t-test. Bars indicate means ± S.E.M.

The following figure supplement is available for figure 9:

**Figure supplement 1.** Depletion of various skin-dLN DC subsets in different DTR mice.

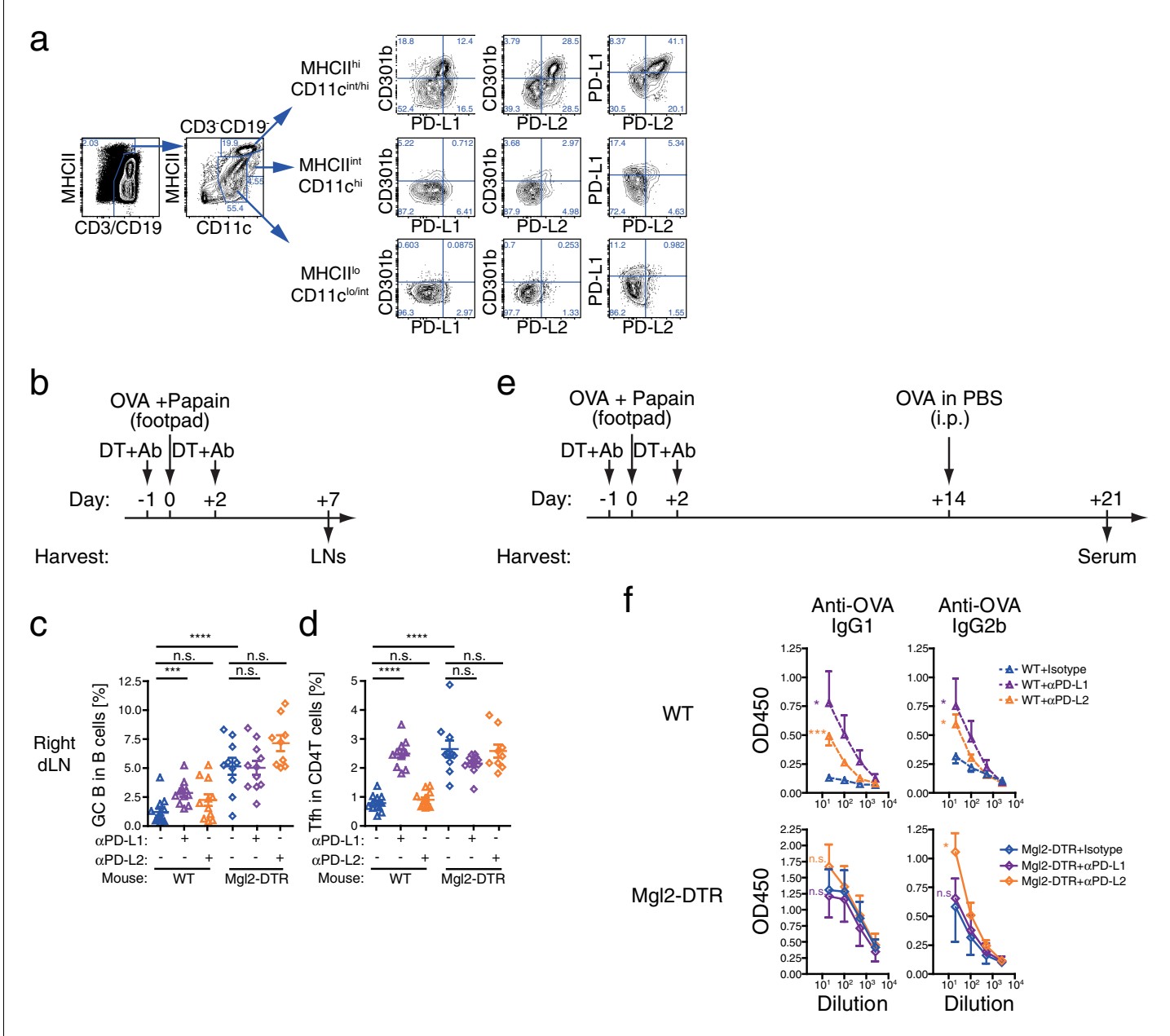

**Figure 10.** Blockade of PD-L1 at the time of immunization enhances Tfh and GC B cell responses in CD301b[+] DC-dependent manner. (a) Pooled skin-dLN cells from naïve WT mice were stained and gated as indicated. Representative flow cytometry plots from three independent experiments are shown. (b–f) WT and Mgl2-DTR mice were injected i.p. with 250 μg anti-PD-L1 mAb (αPD-L1) or αPD-L2 mAb along with 0.5 μg DT on days −1 and +2, then immunized with 50 μg papain and 5 μg OVA in the footpad on day 0. (c,d) Right draining popliteal LNs were collected on day seven and analyzed for GC B cells (c) and Tfh cells (d). Data were pooled from two to three independent experiments. Each dot indicates an individual mouse. (e,f) Alternatively, mice received additional i.p. injection of OVA without adjuvant on day 14, and OVA-specific serum antibody titers were examined on day 21 by ELISA. n.s., not significant, *p<0.05, **p<0.01, ***p<0.001, ****p<0.0001 by two-tailed Student's t-test. Bars indicate means ± S.E.M.

The following figure supplements are available for figure 10:

**Figure supplement 1.** Depletion of CD301b[+] DCs does not reduce PD-1 ligand expression on the residual migratory DCs.

**Figure supplement 2.** Differential effects of PD-1 ligand blockade on CD4T and B cell compartments in CD301b[+] DC-intact and depleted animals.

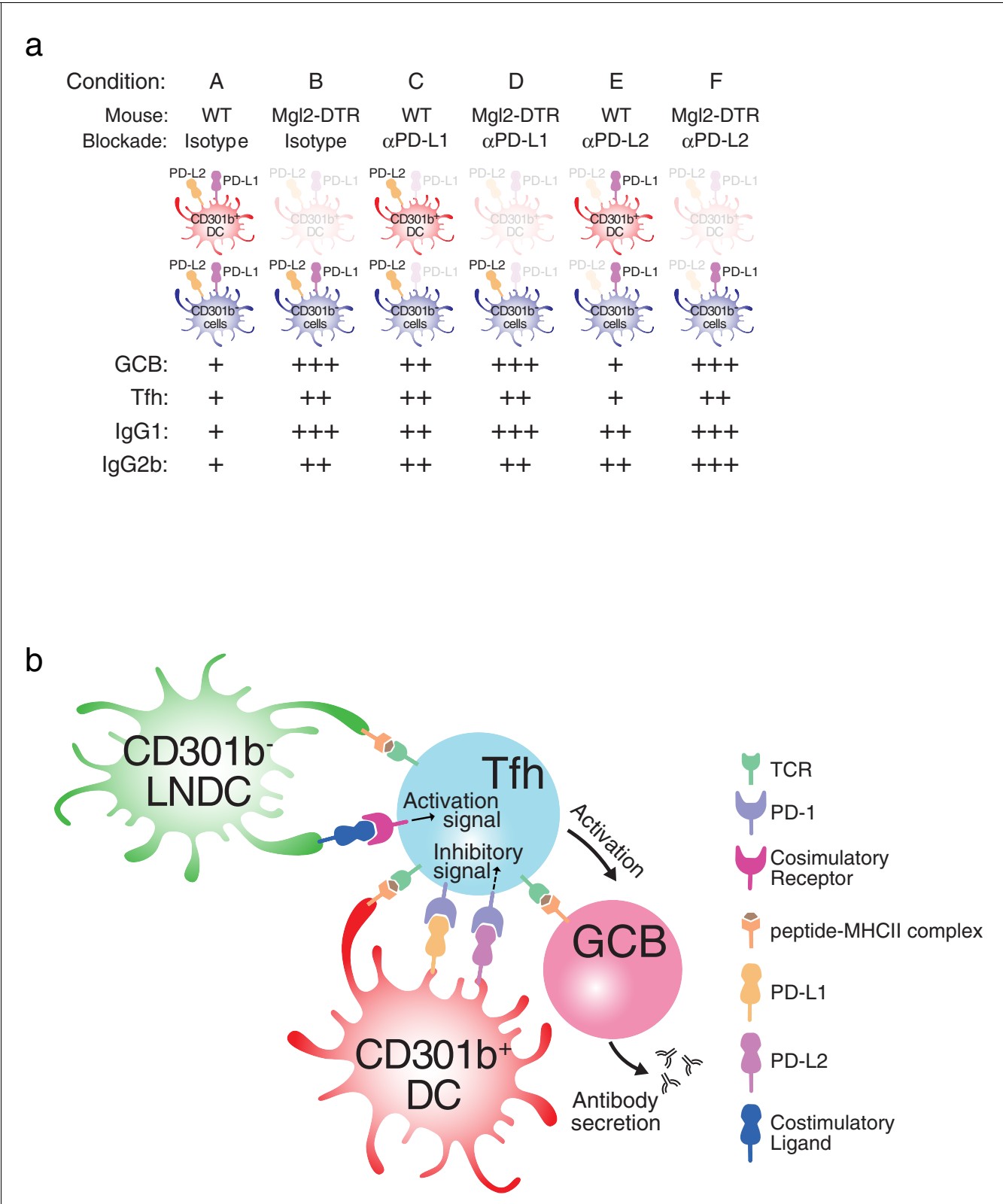

**Figure 11.** Model for CD301b[+] DC-mediated suppression of humoral immunity. (**a**) Summary of the results shown in this study. (**b**) Model for Tfh and GCB cell suppression by CD301b[+] DCs. CD301b[+] DCs express high levels of PD-L1 to inhibit the expansion of Tfh. The impaired Tfh expansion in turn leads to suboptimal GC B cell activation and reduced antibody responses. In the absence of CD301b[+] DCs, CD301b[-]LN-resident DCs present soluble antigens to CD4[+] T cells to induce Tfh differentiation. This CD301b[-]LN-resident DC subset is susceptible to depletion in CD11c-DTR mice.

monoclonal antibodies (mAbs) during the priming phase (*Figure 10b*). We first confirmed that the blockade did not directly deplete CD301b⁺ DCs (*Figure 10—figure supplement 1b*). In the dLN in WT mice, the PD-L1 but not PD-L2 blockade significantly enhanced the GC B cell and Tfh cell development (*Figure 10c,d* and *Figure 10—figure supplement 2a,b*). In Mgl2-DTR mice, however, anti-PD-L1 blocking mAb did not exert further enhancement of the GC B cell and Tfh cell development (*Figure 10c,d* and *Figure 10—figure supplement 2a,b*). Since PD-L1 expression was still intact in the residual DC subsets (*Figure 10—figure supplement 1a*), these data indicate that PD-L1 in the residual cells does not have suppressive function on GCB or Tfh differentiation during priming. In the contralateral ndLNs, neither PD-L1 nor PD-L2 significantly enhanced the GC B cell development in either WT or Mgl2-DTR mice (*Figure 10—figure supplement 2c*).

The blockade of PD-1, PD-L1, or PD-L2 in WT mice during the priming with OVA and papain resulted in significant increases in OVA-specific antibody production (*Figure 10—figure supplement 2d*). Consistent with the GC B cell expansion, antibody responses were enhanced by anti-PD-L1 antibody treatment only in the presence of CD301b⁺ DCs, whereas the PD-L2 blockade further increased OVA-specific IgG2b (*Figure 10e,f*). These data indicate that anti-PD-L1 blocking mAb enhances GC B cell responses in the dLNs by primarily targeting CD301b⁺ DCs in CD301b⁺ DC-intact animals, and suggest that CD301b⁺ DCs use PD-L1 to control Tfh and GC B cell induction. In contrast, the increases in IgG2b responses by the PD-L2 blockade in the absence of CD301b⁺ DCs suggest a role for PD-L2 expressed on CD301b⁻ cells in additionally suppressing humoral immunity. It should be noted, however, that the depletion of CD301b⁺ DCs might have maximized the GCB and antibody responses to the point that further treatment with anti-PD-L1 or PD-L2 mAbs was unable to reveal additional enhancement. Future studies are needed to selectively knockout PD-L1 or PD-L2 from CD301b⁺ DCs in order to clearly address the role of these ligands in suppression of Tfh, GCB and antibody responses.

Altogether, these results support the model that CD301b⁺ DCs provide inhibitory signals at the time of priming of an immune response to control the extent of antibody responses through regulation of Tfh and GC B cell differentiation (*Figure 11*).

## Discussion

In this study, we report regulation of Tfh and antibody responses mediated by the CD301b⁺ DCs. Depletion of CD301b⁺ DC during the first few days of priming leads to an exaggerated Tfh and GC B cell expansion in the dLNs, and a robust increase in systemic isotype switched antigen-specific antibody responses. The enhanced antibody responses did not require any adjuvants, and were observed irrespective of the dose of antigen used. Depletion of CD301b⁺ DCs led to antibodies with high affinity and broader specificity to the immunogen. Depletion of other migrant DC subsets had no impact on antibody responses. Among non-lymphocytes in the LNs, CD301b⁺ DCs expressed the highest levels of PD-1 ligands, PD-L1 and PD-L2. Transient blockade of PD-L1 led to increases in Tfh and GC B cell differentiation in the dLN and enhanced antibody responses, only in the presence of CD301b⁺ DCs. Remarkably, transient depletion of CD301b⁺ DCs led to the generation of autoreactive antibodies. Therefore, our data reveal a novel function of CD301b⁺ DCs in negatively regulating antibody responses and preventing autoantibody formation.

Our results suggest that CD301b⁺ DCs play a central role in regulating humoral immunity by suppressing Tfh ad GC B cell development. With the exact same immunization protocol, previous studies have shown that CD301b⁺ DCs are required for differentiation of antigen-specific CD4⁺ T cells into Th2 cells but not for their differentiation into Th1 or Th17 (*Gao et al., 2013*; *Kashem et al., 2015a*; *Kumamoto et al., 2013*; *Murakami et al., 2013*). Taken together, these results indicate that, Tfh and Th2 responses can be uncoupled, with the CD301b⁺ DCs as the pivotal cell type carrying out suppressive vs. supportive roles, respectively.

Tfh responses are also regulated by the Tfr cells (*Sage et al., 2013*). We describe a separate mechanism dedicated to suppressing Tfh responses mediated by the CD301b⁺ DCs. These non-redundant Tfh regulatory mechanisms likely control Tfh responses at distinct stages. CD301b⁺ DC-mediated suppression occurs within the first few days of primary immune response, while the Tfr entry into the reactive LN occurs after 7 days of immunization (*Sage et al., 2013*). These two phases of Tfh regulation likely provide complementary mechanisms of Tfh control. For instance, Tfr cells are important in producing high affinity antibody responses that are devoid of self-reactivity

(*Pratama and Vinuesa, 2014*). Regulation of Tfh responses at early phases by CD301b$^+$ DCs may be critical in preventing excess antibody responses to foreign and self-antigens, which can lead to host pathology through complement fixation, and deposition of antibody-immune complex (*Rouse and Sehrawat, 2010*). In fact, we detected antinuclear autoantibodies following transient depletion of CD301b$^+$ DCs. Future studies are needed to determine the consequence of CD301b$^+$ DC depletion on the development of autoimmune disease.

CD301b$^+$ DCs are the major subset of CD11b$^+$ migratory DCs that most efficiently take up the antigen and bring to the dLNs within one day after immunization with papain. In addition, CD301b$^+$ DCs are required for optimal screening and activation of naïve CD4$^+$ T cells within the dLNs (*Kumamoto et al., 2013*). Our current study showed that, despite the reduction in the amount of antigen available within the dLNs and the impaired recruitment of naïve lymphocytes to the dLN in the first three days of immunization (*Kumamoto et al., 2013*), development of Tfh and GC B cells was amplified in the absence of CD301b$^+$ DCs. Furthermore, depletion of CD301b$^+$ DCs led to antibody production to a soluble antigen even in the absence of adjuvants. Specifically, depletion of CD301b$^+$ DCs lifted the suppression on Tfh such that their numbers and percentages increased within the LNs, leading to preferential activation of isotype switched antibody-secreting cells. It is interesting, however, that antibody responses against *N. brasiliensis* in CD301b$^+$ DC-depleted animals are comparable to WT animals (*Kumamoto et al., 2013*). Similarly, CD301b$^+$ DCs did not suppress IgG2b antibody generated by strong type 1 immunity induced by high doses of CpG (*Figure 4c*). These results suggest that pathogens can overcome CD301b$^+$ DC-dependent suppression of antibody responses, likely through continuous antigen presentation (*Baumjohann et al., 2013*; *Deenick et al., 2010*) and through engagement of strong inductive signals through pattern recognition receptors including TLRs (*Pasare and Medzhitov, 2005*).

Our study pointed to the role of LN-resident DCs in supporting Tfh-inductive functions, as evidenced by the diminished antibody responses in CD11c-DTR;Mgl2DTR mice but not those depleted of migrant DCs. The DC subset specifically responsible for inducing Tfh remains to be identified. Alternatively, our results suggest the possibility that any DC subset is sufficient to induce Tfh cells in the absence of the negative regulation exerted by CD301b$^+$ DCs (*Lahoud et al., 2011*; *Yao et al., 2015*).

Recent studies have indicated that the therapeutic effect of PD-L1 blockade in chronic infection by HIV and malaria is mediated in part by inducing more Tfh cells that help B cells to produce pathogen-specific neutralizing antibodies (*Butler et al., 2012*; *Cubas et al., 2013*). Similarly, continuous blockade as well as genetic deletion of PD-L1 was shown to enhance Tfh and GC B cell expansion upon type 2 immunization (*Hams et al., 2011*). In these studies, due to the timing of the PD-L1 blockade (past Tfh priming), activated B cells were the major source for the PD-L1 relevant to the memory Tfh population. In contrast, our results clearly demonstrated that the transient depletion of CD301b$^+$ DCs during the priming phase had a significant positive impact on humoral immunity that persisted even after the secondary immunization. Our study shows that the PD-L1 blockade cannot further enhance Tfh and GC B responses when CD301b$^+$ DCs are depleted. Consistently, a recent report demonstrated that neonatal blockade of PD-L1 that is preferentially expressed by CD11b$^+$ DCs in the lung results in exaggerated antibody production later in life during adulthood, suggesting the contribution of a similar DC subset that suppresses humoral immunity through PD-L1 expression (*Gollwitzer et al., 2014*). As many other cell types express PD-L1 in vivo, dissection of the relative and temporal contributions of PD-L1 in Tfh blockade by specific cell types will require generation of new tools.

Our results are consistent with an inhibitory role for PD-L1 and/or PD-L2 for antibody responses described in some studies (*Hams et al., 2011*; *Ishiwata et al., 2010*), but not in others (*Good-Jacobson et al., 2010*; *Sage et al., 2013*). For instance, GC B cell and IgG1 induction in response to NP-CGG immunization in PD-1 knockout mice are comparable or even slightly better than WT mice (*Good-Jacobson et al., 2010*), consistent with our data. In contrast, the PD-1 deficient mice contain higher percentages of Tfr but not Tfh (*Sage et al., 2013*), and Tfh cells isolated from PD-1 deficient mice had impaired cytokine responses (*Good-Jacobson et al., 2010*). Differences observed in our study versus others using genetically deficient mice might be explained by the fact that we used transient blockade of PD-1 ligands with mAbs. However, since PD-1-independent mechanism is also possible, the precise mechanism by which CD301b$^+$ DCs regulate GC B needs to be elucidated in future studies.

Autoreactive antibody responses are controlled at the level of the bone marrow in developing B cells (central tolerance) as well as at the level of the peripheral tissues (peripheral tolerance). Autoreactive early immature B cells recognizing self-antigens are eliminated in the BM in a cell-intrinsic manner. Once in the periphery, mature B cells that react against autoantigens are suppressed by regulatory T cells (Tregs) in a cell-extrinsic manner (*Meffre and Wardemann, 2008*). In the current study, we show an additional mechanism for keeping autoreactive antibodies under control that is mediated by the CD301b$^+$ DCs. Based on our analysis of protein antigen immunization in the absence of adjuvants, we believe that the loss of B cell tolerance may be due to the emergence of self-reactive Tfh cells in CD301b$^+$ DC-depleted mice. Future studies are needed to determine whether such self-reactive Tfh cells are normally suppressed by CD301b$^+$ DCs through direct interaction via PD-1:PD-L1, or through an indirect pathway involving Tregs, and whether depletion of other DC subsets results in the development of autoreactive antibodies.

The results of our current study have important implications on vaccine designs and on the treatment of autoimmune diseases. For example, autoimmune diseases driven by hyperstimulation of B cells, such as SLE, might benefit from enhancing the function of CD301b$^+$ DCs and its suppressive mechanisms. For such therapy to be efficacious, CD301b$^+$ DCs must be able to reverse a preexisting state of B cell stimulation. On the other hand, inhibiting the functions of CD301b$^+$ DCs and/or PD-1 ligands at the time of vaccination may improve the efficacy of a suboptimal vaccine to generate robust isotype-switched immunoglobulin responses. Our data showed that PD-L1 blockade could be delivered transiently at the time of vaccination, alleviating concerns of untoward consequences of long term inhibition of PD-1 engagement (*Brahmer et al., 2012*; *Topalian et al., 2012*). While PD-L1 blockade with an antibody is not a practical solution to boost prophylactic vaccine approaches due to the inhibitory costs associated with the antibody, future development of an affordable PD-L1 antagonist might enable implementation of this concept into vaccine design. An optimal combination of the adjuvant to activate B cell affinity maturation and a blocker of inhibition imposed by the CD301b$^+$ DC may provide an ideal setting for generating robust protective humoral responses.

## Materials and methods

### Mice

WT C57BL/6N (B6) and CD45.1 (B6.SJL-*Ptprc$^a$Pepc$^b$*/BoyJ) OT-II mice were maintained in our specific pathogen-free facility. *Mgl2$^{DTReGFP/DTReGFP}$* mice (RRID:MGI:5510785) were developed as previously described (*Kumamoto et al., 2013*) and bred with B6 mice to gain heterozygotic Mgl2-DTR mice (RRID:MGI:5510786). huLangerin-DTR mice were bred with B6 or *Mgl2$^{DTReGFP/DTReGFP}$* homozygotes. Transgene-bearing offspring were screened as previously described (*Bobr et al., 2010*). msLangerin-DTR mice (B6.129S2-*Cd207$^{tm3(DTR/GFP)Mal}$*/J; RRID:IMSR_JAX:016940) were purchased from the Jackson Laboratory (Bar Harbor, ME) and maintained as homozygotes. For depletion experiments, the homozygotic msLangerin-DTR mice were crossed onto *Mgl2$^{DTReGFP/DTReGFP}$* homozygotes to gain heterozygotic msLangein-DTR;Mgl2-DTR (*Mgl2$^{+/DTReGFP}$*;*Cd207$^{+/DTRGFP}$*) offspring. To generate CD11c-DTR;Mgl2-DTR mice, CD11c-DTR mice (B6.FVB-Tg(*Itgax*-DTR/EGFP)57/Lan/J; RRID:IMSR_JAX:004509) were crossed with *Mgl2$^{DTReGFP/DTReGFP}$* homozygotes, then transgene-bearing offspring were screened with the following primers to amplify the boundary between the *Itgax* promoter and the DTReGFP cassette: CAGAGCCTGCTTCTGTTCTCCAG and GTTGCTGGTTCCAGCAGCTAG. For generating chimeric mice, WT B6 mice were irradiated with two doses of 475 cGy (total 950 cGy) and reconstituted with 1 x 10$^6$ WT, Mgl2-DTR or 1:1 mixture of WT and Mgl2-DTR BM cells. The mice were used for experiments more than eight weeks after the BM transplantation. In most experiments, 2–6 month-old female mice were used. However, due to mouse availability, mice of both sexes were used in experiments involving mice with huLangerin-DTR or CD11c-DTR construct. No gross difference was noted between male and female in those experiments. All animal protocols (protocol number 10365) were approved by and performed in accordance with guidelines set by the Institutional Animal Care and Use Committee at Yale University.

### Immunization

DT or its non-toxic mutant CRM197 (List Biological Laboratories, Campbell, CA) was injected i.p. (0.5 µg in 0.5 ml PBS per dose, except for CD11c-DTR;Mgl2-DTR mice in which only a single dose of

125 ng per mouse was given to avoid toxicity) at indicated time-points. For antibody titer analyses, mice were immunized in the hind footpad on day 0 with indicated dose of OVA (Ovalbumin Low Endo, Worthington Biochemical Corporation, Lakewood, NJ) with 50 µg papain (Sigma-Aldrich, St. Louis, MO), CpG-2216 (TriLink Biotechnologies, San Diego, CA), or LPS (from *Salmonella enterica* serotype typhimurium, Sigma). Alternatively, mice were immunized i.p. with 10 µg OVA suspended in 50% alum in PBS (Imject Alum, Thermo Scientific, Waltham, MA). The immunization was boosted on day 14 by injecting 10 µg OVA in 100 µl PBS i.p. or retro-orbitally without adjuvant. In some experiments, $NP_{16}OVA$ (Biosearch Technologies, Petaluma, CA) was used for immunization. In experiments involving BM chimeric mice, an additional boost was given on day 21. For LN cell analyses, mice were immunized on day 0 with 5 µg OVA plus 50 µg papain in 20 µl PBS in the right hind footpad and popliteal LNs were harvested at indicated time-points. Where indicated, TCR-transgenic $CD4^+$ T cells were isolated from the spleen and LNs of naïve CD45.1 ($Ptprc^a$)-expressingOT-II mice by $CD4^+$T cell isolation kit (Miltenyi Biotec, San Diego, CA) and transferred (5–10 × $10^5$ cells per recipient) retro-orbitally 4–6 hr before the first DT treatment. For antibody treatments, mice were injected i.p. at indicated time-points with 50 µg per dose anti-Gr-1 (clone RB6-8C5, BioLegend, San Diego, CA) or rat IgG2b isotype control (RTK4530, BioLegend), or with 250 µg per dose anti-PD-1 (clone RMP1-14, BioXCell, West Lebanon, NH), anti-PD-L1 (clone 10F.9G2, BioXCell) or anti-PD-L2 (clone TY25, BioXCell).

## Flow cytometry

LNs and spleen were minced and digested with collagenase D (Roche, Indianapolis, IN) for 30 min. Peritoneal exudates were collecte by flushing with PBS. Dead cells were gated out by staining with LIVE/DEAD Fixable Aqua Dead Cell Stain Kit (Invitrogen, Carlsbad, CA). For staining T and B cell subsets, cell surface markers were stained with mAbs against CD4 (GK1.5), CD19 (6D5), CD45.1 (A20), IgD (11-26c.2a), CD38 (90), CD138 (281–2), PD-1 (RMP1-30), CD44 (IM-7), GL7 (GL7), ICOS (C398.4A), CXCR5 (2G8, BD Pahrmingen, San Diego, CA), CD95 (Jo2, BD Pharmingen) and/or PSGL1 (2PH1, BD Pharmingen). For staining DC subsets, cells were stained with mAbs against CD3 (17A2), CD19 (6D5), MHCII (M5/114.15.2), CD11c (N418), CD11b (M1/70), CD24 (M1/69), CD64 (X54-5/7.1), Ly6C (HK1.4), Ly6G (1A8), CD103 (2E7), CD326 (G8.8), PD-L1 (MIH5, eBioscience, San Diego, CA), PD-L2 (TY25), and/or CD301b (11A10-B7-2) (*Kumamoto et al., 2013*). For transcriptional factor staining, cells were fixed with Foxp3/Transcription Factor Fixation/Permeabilization Concentrate and Diluent (eBioscience) and stained for Zbtb46 (U4-1374, BD Pharmingen), Bcl6 (BCL-DWN, eBioscience) or Foxp3 (FJK-16s, eBioscience). For intracellular cytokine staining, LN cells were stimulated with Cell Stimulation Cocktail (eBioscience) for an hour at 37°C, then for another 4 hr with Protein Transport Inhibitor Cocktail (eBioscience). Cells were then fixed with Cytofix/Cytoperm Fixation/Permeabilization Solution (BD Biosciences, San Jose, CA) and stained for IL-4 (11B11) and IL-21 (recombinant mouse IL-21R Fc chimera protein, R&D Systems, Minneapolis, MN). Gating was set based on the binding of relevant non-specific antibodies or human Fc chimera protein (R&D Systems). For detecting OVA-specific B cells, surface-stained cells were fixed and permeabilized with BD Cytofix/Cytoperm, then incubated with 10 µg/ml Alexa Fluor 488-conjugated OVA (A488-OVA) for 30 min on ice. All mAbs were purchased from BioLegend unless otherwise indicated. Stained cells were scanned on BD FACS LSR II (BD Biosciences) and analyzed by FlowJo software (Version 9.3.2, TreeStar, Ashland, OR).

## ELISA

OVA-specific serum antibody titers were measured by a standard ELISA protocol. Briefly, MaxiSorp ELISA plates (Thermo Scientific) were coated with 10 µg/ml OVA in carbonate buffer. For affinity maturation analyses, additional plates were coated with 10 µg/ml $NP_{16}OVA$, $NP_4BSA$, or $NP_{34}BSA$ (Biosearch Technologies). Plates were then incubated with PBS containing 5% fetal bovine serum to block non-specific bindings. Sera were diluted in the blocking buffer as indicated, and applied to the plates. Subclass-specific antibodies that bound to OVA were detected by horseradish peroxidase (HRP)-conjugated goat anti-mouse IgM (Southern Biotech), IgG1, IgG2b or subclass-nonspecific IgG (Jackson ImmunoResearch Laboratories, West Grove, PA) or by biotinylated anti-mouse IgE (RME-1, BioLegend) followed by HRP-streptavidin. Total serum immunoglobulin content was determined with ELISA plates coated with 10 µg/ml goat anti-mouse IgG+M (H+L) (Jackson ImmunoResearch),

followed by incubation with HRP-conjugated subclass-specific detection antibodies. Serum anti-nuclear IgG antibodies were captured by Mouse ANA Total Ig ELISA Kit (Alpha Diagnostics, San Antonio, TX) and detected by HRP-conjugated goat anti-mouse IgG Fcg Fragment (Jackson Immu-noResearch). HRP binding was visualized by 1xTMB ELISA Substrate Solution (eBioscience) and the absorbance at 450 nm was detected by a plate reader (Model 680, Bio-Rad, Hercules, CA). For calculating $EC_{50}$ and the maximal absorbance ($A_{max}$), data from each mouse were fit to $Y = A_{max}*X/(EC_{50}+X) + NS*X + Background$, where Y is observed absorbance and X is serum dilution, with the following restraints: $0 < A_{max} < 3.5$, $0 < EC_{50} < 1$, $0 < NS < 2$, and $0 < Background < 0.1$. $EC_{50}$ was assumed as 1.0 where calculated $A_{max}$ was < 0.2. Affinity maturation was examined by taking the ratio between $EC_{50}$ for $NP_4BSA$ binding and $EC_{50}$ for $NP_{34}BSA$ binding and expressed as the negative log value of the ratio.

### Antibody responses against house dust mite (HDM)

Crude lysates of *D. pteronyssinus* were purchased from Greer Laboratories (Lenoir, NC). For sensitization, about 1x2 cm$^2$ area of upper back skin was shaved with a razor on day $-1$ and painted with 80 µg (as protein concentration in 0.1 ml PBS) of lysates on days 0, 7 and 14. HDM-specific IgG1 and IgG2b were measured in the day 20 sera by ELISA using plates coated with HDM lysates. For detecting HDM-specific IgE, plates coated with anti-mouse IgE (R35-72, BD Biosciences) were incubated with the sera, then further incubated with biotinylated HDM lysates.

### Statistics

All statistical analyses were performed by two-tailed Student's t-test with Welch's correction and all bars in graphs indicate means ± S.E.M., unless otherwise indicated. n.s., not significant ($p \geq 0.05$), *$p < 0.05$, **$p < 0.01$, ***$p < 0.001$, ****$p < 0.0001$.

## Acknowledgement

We thank R Medzhitov for critical discussion, and H Dong for technical assistance. Supported by the US National Institutes of Health (AI054359 and AI062428 to AI, R01-AR067187 to DHK). This work was supported in part by grants provided by AbbVie. The study was partly funded by a pilot grant from the Yale Rheumatic Diseases Research Core Center, NIH (NIAMS) P30 AR053495. YK was an Astellas Foundation for Research on Metabolic Disorders fellow. TH is a Japan Society for the Promotion of Science Postdoctoral Fellow for Research Abroad. PWW held a Gruber Science Fellowship and is supported by the National Science Foundation Graduate Research Fellowship Program. AI is an investigator of the Howard Hughes Medical Institute. The authors have no conflicting financial interests.

## Additional information

### Funding

| Funder | Grant reference number | Author |
| --- | --- | --- |
| National Institutes of Health | AI054359 | Akiko Iwasaki |
| National Institutes of Health | AI062428 | Akiko Iwasaki |
| National Institutes of Health | R01-AR067187 | Daniel H Kaplan |
| National Institute of Arthritis and Musculoskeletal and Skin Diseases | P30 AR053495 | Akiko Iwasaki |
| Astellas Foundation for Research on Metabolic Disorders | Metabolic Disorders fellow | Yosuke Kumamoto |
| AbbVie | AbbVie-Yale Collaboration | Akiko Iwasaki |

The funders had no role in study design, data collection and interpretation, or the decision to submit the work for publication.

## Author contributions

YK, Conception and design, Acquisition of data, Analysis and interpretation of data, Drafting or revising the article; TH, Acquisition of data, Analysis and interpretation of data, Contributed unpublished essential data or reagents; PWW, Acquisition of data, Analysis and interpretation of data; DHK, Analysis and interpretation of data, Drafting or revising the article, Contributed unpublished essential data or reagents; AI, Conception and design, Analysis and interpretation of data, Drafting or revising the article

## Author ORCIDs

Akiko Iwasaki, http://orcid.org/0000-0002-7824-9856

## Ethics

Animal experimentation: All animal protocols were approved by and performed in accordance with guidelines set by the Institutional Animal Care and Use Committee at Yale University. (protocol number 10365)

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
