## [Decision Letter]

Thank you for submitting your article "CD301b^+^ dendritic cells suppress humoral immunity against protein antigens" for consideration by *eLife*. Your article has been reviewed by three peer reviewers, and the evaluation has been overseen by a Reviewing Editor and Tadatsugu Taniguchi as the Senior Editor. The reviewers have opted to remain anonymous.

The reviewers have discussed the reviews with one another and the Reviewing Editor has drafted this decision to help you prepare a revised submission.

All three referees found the data showing substantial increases in Tfh cellularity, GC B cell generation and antibody production following depletion of CD301b^+^ DC and after immunization with a diverse set of antigens and adjuvants to be very intriguing and potentially of high significance. The existence of a cellular mechanism that actively suppresses the production of a high-titer/high-affinity antibody response is clearly an interesting finding. However, several important concerns were raised that need to be addressed in a revised paper before the work can be accepted for publication in *eLife*.

Essential issues that must be addressed:

1) A key issue is connecting and reconciling the current data with an earlier study by the same authors (Kumamoto, Immunity 2013). In this earlier manuscript the authors described a nearly complete abolishment of IL-4 production by CD4 T cells in LN and spleens after CD301b^+^ DC depletion and OVA+papain immunization. Given that IgE and IgG1 antibody class-switching is generally accepted to be IL-4 dependent, the current description of substantial increases in such antibody isotypes is difficult to reconcile with the previously published data. A more substantial effort is thus required to consolidate the two findings and explain how one generates type-2 antibody responses in absence of type-2 cellular immunity. For example, is there a substantial increase in IL-4 production from non-T cell sources (i.e. basophils, mast cells) after DC depletion? Is there an increase in IL-21 production that is capable of substituting for lack of IL-4? Are other non type-2 antibodies, such as IgG2a, IgG3 also increased? The 2013 manuscript demonstrated substantial increases in frequency of IFNγ producing cells, which would in principle skew the humoral response towards these antibody subclasses. In this regard, it would be helpful to demonstrate intracellular IL-4 and IFNγ cytokine levels after ex vivo restimulation on the non-Tfh, Tfh and GC-Tfh phenotype CD4 T cells following immunization in this depletion model.

2) The authors go through considerable effort to show that this phenomenon is specific to depletion of the CD301b^+^ DC population, and is not a simple artifact of cell death due to DT treatment. However, a possibility remains that modulation of Ab production in these mice is due to an off target effect of DT injection through the depletion of a DC-independent Mgl2+ population. Can the authors provide a detailed description of the composition of macrophages and DC in the skin DLN and non-DLN and peritoneal cavity of Mgl2+-DTR mice treated with DT or PBS? These data should be added to Figure 2. It would also be instructive to compare directly ex-vivo the capacity of OVA-pulsed Mgl2+ and Mgl2- cutaneous DC to promote Tfh differentiation and expansion.

3) Technically, the least convincing section of the manuscript is the attribution of DC inhibition of antibody responses to the PD-1 pathway. This is based primarily on the proposed absence of an additive effect between CD301b^+^ DC depletion and anti-PDL1/2 treatment. This is somewhat problematic, for the following reasons:

First, although anti-PDL-1 does not appear to have an additive effect over DC depletion, Figure 9 show increases in GC/antibody responses with anti-PDL2 treatment (although the comparison in 9C does not reach statistical significance with the small N provided, but this does not mean that there's no difference).

Second, how sure are the authors that anti-PDL1 antibody treatment does not lead to depletion of CD301+DCs? Is this population still there after treatment? Also, do the authors obtain the same effects by blocking with an anti-PD-1 antibody? Anti PD-1 treatment is mentioned in the methods but does not seem to be discussed in the main text.

Third, even if PD-L1/2 antibodies do not deplete CD301b^+^ DCs, the absence of an additive effect for blocking PD-1 ligands does not necessarily mean this is their mechanism of action – it could well be that DC depletion alone already increases the response to a level that is too high for any effects of PD-1 ligand blocking to be noticeable.

Here again assessing the contribution of PDL-1 and PDL1 to Mgl2+ DC modulation of Tfh expansion ex vivo should be informative.

[Editors' note: further revisions were requested prior to acceptance, as described below.]

Thank you for resubmitting your work entitled "CCD301b^+^ dendritic cells suppress T follicular helper cells and antibody responses to protein antigens" for further consideration at *eLife*. Your revised article has been favorably evaluated by Tadatsugu Taniguchi (Senior editor), a Reviewing editor, and three reviewers.

The manuscript has been improved but there are some remaining issues that need to be addressed before acceptance, as outlined below:

1) The new dataset in Figure 8 demonstrating increased frequency of IL-4/IL-21-producing TfH should show how the placement of the gates for cytokine-positive cells were determined (isotype or naive control plots would help).

2) The new dataset on autoantibody production presented in Figure 6 is very interesting. It would be helpful to know whether such effects are specifically due to the CD301b^+^ DC depletion or to DC death in general, and would be observed in a DT-treated Langerin-DTR system, as an example.

3) Please indicate whether the OVA used for these studies was endotoxin-free. Although not significantly altering the major findings of the manuscript, this would be important for the conclusions reached for the adjuvant-free studies.

4) The finding that DT injection leads to substantial accumulation of Ly6C^+^ cells in the peritoneum suggests that DT leads to inflammatory events that could have triggered increased myelopoiesis or promoted the release of myeloid cells which in turn could have contributed to the dysregulated immune response. This should be mentioned by the authors in the Discussion.

5) The authors have not addressed concerns regarding the contribution of PDL1 in CD103b^+^ DC-mediated control of Tfh and B cell response. Please remove strong claims regarding the potential role of PDL-1 from the Abstract and modulate their claim in this regard in the Results and Discussion sections.

---

## [Author Response]

Essential issues that must be addressed:

1) A key issue is connecting and reconciling the current data with an earlier study by the same authors (Kumamoto, Immunity 2013). In this earlier manuscript the authors described a nearly complete abolishment of IL-4 production by CD4 T cells in LN and spleens after CD301b^+^ DC depletion and OVA+papain immunization. Given that IgE and IgG1 antibody class-switching is generally accepted to be IL-4 dependent, the current description of substantial increases in such antibody isotypes is difficult to reconcile with the previously published data. A more substantial effort is thus required to consolidate the two findings and explain how one generates type-2 antibody responses in absence of type-2 cellular immunity. For example, is there a substantial increase in IL-4 production from non-T cell sources (i.e. basophils, mast cells) after DC depletion?

We agree with the referees that it is important to determine how so called “type 2” antibodies are being generated in the absence of Th2 cells.

We observed a slight increase in IL-4 production by non-T cells, though, given their low frequency above the background, we are hesitant to be conclusive and do not wish to include the data for publication (see Figure 12). Instead, to our excitement, even though the Th2 responses was reduced (consistent with our study in 2013), we found that the Tfh cells generated in the absence of CD301b DCs produce intact levels of IL-4 and IL-21 (comparable to the undepleted mice). In addition, there were a lot more Tfh cells in the CD301b-depleted mice. We believe that the combination of the increase in the number of Tfh, as well as their robust secretion of IL-4 and IL-21, explains why the CD301b-depleted mice are capable of generating the type 2 antibody responses including IgE and IgG1. These data are included as new Figure 8.

Author response image 1.IL-4 production from non-T cells in CD301b^+^ DC-depleted mice.Dashed line indicates staining background.**DOI:**
http://dx.doi.org/10.7554/eLife.17979.019

Is there an increase in IL-21 production that is capable of substituting for lack of IL-4?

As stated above, we examined the production of IL-21 and IL-4 in CD301b-depleted or intact mice and observed intact production of IL-21 and IL-4 in in antigen-specific Tfh cells. These data are now included in Figure 8.

Are other non type-2 antibodies, such as IgG2a, IgG3 also increased?

We measured IgG3 levels in CD301b-depleted immunized animals. IgG3 responses were not induced upon immunization with papain + OVA. Please see Figure 13).

Author response image 2.IgG3 production in mice immunized with OVA plus papain in the footpad.All mice received intraperitoneal injection of OVA without papain on day 14.**DOI:**
http://dx.doi.org/10.7554/eLife.17979.020

*The 2013 manuscript demonstrated substantial increases in frequency of IFNγ producing cells, which would in principle skew the humoral response towards these antibody subclasses. In this regard, it would be helpful to demonstrate intracellular IL-4 and IFNγ cytokine levels after* ex vivo *restimulation on the non-Tfh, Tfh and GC-Tfh phenotype CD4 T cells following immunization in this depletion model.*

In our 2013 manuscript in Immunity, we demonstrated a trend for increase in IFN-g producing cells (however not significant).

As described above, we demonstrate that the OVA-specific CXCR5^-^ OT-II cells (Th2 cells) indeed express lower levels of IL-4, whereas CXCR5^+^OT-II Tfh cells express intact levels of IL-4 and IL-21, and are much more numerous in the CD301b-depleted mice (see our new data Figure 8). We believe this explains why the humoral responses are skewed towards the “type 2” antibody classes.

2) The authors go through considerable effort to show that this phenomenon is specific to depletion of the CD301b^+^ DC population, and is not a simple artifact of cell death due to DT treatment. However, a possibility remains that modulation of Ab production in these mice is due to an off target effect of DT injection through the depletion of a DC-independent Mgl2+ population. Can the authors provide a detailed description of the composition of macrophages and DC in the skin DLN and non-DLN and peritoneal cavity of Mgl2+-DTR mice treated with DT or PBS? These data should be added to Figure 2.

We have examined the macrophage and DC composition in the skin dLN and ndLN and peritoneal cavity of the Mgl2-DTR mice treated with DT or PBS. These results showed a clear depletion of the CD301b^+^ DCs in DT treated mice, and an increase in the Ly6C^+^ cells in the peritoneum. We also added new data on the phenotype of CD301b^+^ cells in the LNs and the peritoneal cavity, showing that they express a conventional DC-specific transcription factor Zbtb46. We have now added these results to Figure 2—figure supplement 1.

It would also be instructive to compare directly ex-vivo the capacity of OVA-pulsed Mgl2+ and Mgl2- cutaneous DC to promote Tfh differentiation and expansion.

We have attempted, several times, to establish an in vitro co-culture system to promote Tfh differentiation and expansion. There is one paper published Shin et al. (Cell Rep, 11:1929), which shows that Tfh can be generated in vitro. We have tried to reproduce their data and to adopt this system to study Tfh in vitro but were unable to do so. Ours is not an isolated failure, as there have never been a reported successful in vitro Tfh culture or DC-Tfh differentiation system using mice. For us to develop a de novo DC-Tfh culture system would require a lot of resource and time, and we consider such endeavor to be a separate project from this current study.

3) Technically, the least convincing section of the manuscript is the attribution of DC inhibition of antibody responses to the PD-1 pathway. This is based primarily on the proposed absence of an additive effect between CD301b^+^ DC depletion and anti-PDL1/2 treatment. This is somewhat problematic, for the following reasons:

First, although anti-PDL-1 does not appear to have an additive effect over DC depletion, Figure 9 show increases in GC/antibody responses with anti-PDL2 treatment (although the comparison in 9C does not reach statistical significance with the small N provided, but this does not mean that there's no difference).

We agree that the GC B cell frequency is slightly elevated in couple of mice in the group that were CD301b^+^ DC-depleted and treated with anti-PDL2 antibody. However, the differences were not statistically significant (Figure 10). As the referees point out, there is a slightly elevated IgG2b, but not IgG1, in anti-PDL2 treated CD301b^+^ DC-depleted animals shown in Figure 10. Since PDL2 expression on the remainingDCs was not affected by the depletion of CD301b^+^ DCs (Figure 10—figure supplement 1), these data suggest that the residual PDL2+ cells play suppressive role on IgG2b production and possibly on GC B cell expansion in a PDL2-dependent manner. We now discuss these possibilities in the second paragraph of the subsection “PD-L1 blockade enhances expansion of Tfh and GC B cells only in the presence of CD301b^+^ 328 DCs”.

Second, how sure are the authors that anti-PDL1 antibody treatment does not lead to depletion of CD301+DCs? Is this population still there after treatment? Also, do the authors obtain the same effects by blocking with an anti-PD-1 antibody? Anti PD-1 treatment is mentioned in the methods but does not seem to be discussed in the main text.

We have examined whether anti-PDL1 or anti-PDL2 treatment depletes CD301b^+^ DCs. Our data clearly show that neither of these antibodies led to the depletion of the CD301b^+^ DCs. We have included these data in Figure 10—figure supplement 1. We also have examined whether blocking PD-1 in WT mice mimics the enhanced antibody production observed in Mgl2-DTR mice. Although previous reports show controversial results on the role of PD-1 in humoral immunity (discussed in the second paragraph of the subsection “PD-L1 blockade enhances expansion of Tfh and GC B cells only in the presence of CD301b^+^ 328 DCs”), the PD-1 blockade during the priming in our immunization protocol indeed showed enhanced antibody production similar to that observed in Mgl2-DTR mice. The data are now shown in Figure 10—figure supplement 2.

Third, even if PD-L1/2 antibodies do not deplete CD301b^+^ DCs, the absence of an additive effect for blocking PD-1 ligands does not necessarily mean this is their mechanism of action – it could well be that DC depletion alone already increases the response to a level that is too high for any effects of PD-1 ligand blocking to be noticeable.

Since there are many PD-L1-expressing cells left intact in CD301b^+^ DC-depleted mice (Figure 10—figure supplement 1), our data indicate that blocking those residual PD-L1 epitopes does not further increase GC B cell responses In addition, we observed a further increase in IgG2b (Figure 10) in CD301b^+^DC-depleted animals treated with anti-PD-L2, the data argue against the idea that GC responses were simply maxed out by the depletion of CD301b^+^ DCs alone. As the referees suggest, however, that the depletion of CD301b^+^ DCs might have elevated the GCB and antibody responses to the point that further treatment with anti-PD-L1 or PD-L2 was unable to reveal additional enhancement, even if they existed. Future studies are needed to selectively knockout PD-L1 or PD-L2 from CD301b^+^ DCs in order to clearly address the role of these ligands in suppression of Tfh, GCB and antibody responses. We now included this discussion in the second paragraph of the subsection “PD-L1 blockade enhances expansion of Tfh and GC B cells only in the presence of CD301b^+^ 328 DCs”.

*Here again assessing the contribution of PDL-1 and PDL1 to Mgl2+ DC modulation of Tfh expansion* ex vivo *should be informative.*

Please see our discussion about ex vivo Tfh induction above.

[Editors' note: further revisions were requested prior to acceptance, as *described below.]*

1) The new dataset in Figure 8 demonstrating increased frequency of IL-4/IL-21-producing TfH should show how the placement of the gates for cytokine-positive cells were determined (isotype or naive control plots would help).

We have now added isotype staining control panels for IL-4 and IL-21 staining in Figure 8.

2) The new dataset on autoantibody production presented in Figure 6 is very interesting. It would be helpful to know whether such effects are specifically due to the CD301b^+^ DC depletion or to DC death in general, and would be observed in a DT-treated Langerin-DTR system, as an example.

While we understand reviewer's concern regarding the potentially non-specific impacts from DC death on autoantibody production, repeating the experiment in Langerin-DTR mice alone will take several more months. Since we show subset specificity of the increased antibodies in the immunization-induced model (Figure 9), we hope that the referees and the editors would agree to let us proceed without having to include the autoantibody experiment in Langerin-DTR mice. Instead, we now inserted a sentence regarding this issue in the Discussion. However, if the editors deem this point to be absolutely critical to our study, we are open to conducting such experiments.

3) Please indicate whether the OVA used for these studies was endotoxin-free. Although not significantly altering the major findings of the manuscript, this would be important for the conclusions reached for the adjuvant-free studies.

We use low-endotoxin OVA (≤1EU/mg) from Worthington. It is now mentioned in the Immunization section in the Methods.

4) The finding that DT injection leads to substantial accumulation of Ly6C^+^ cells in the peritoneum suggests that DT leads to inflammatory events that could have triggered increased myelopoiesis or promoted the release of myeloid cells which in turn could have contributed to the dysregulated immune response. This should be mentioned by the authors in the Discussion.

Although we cannot eliminate possibilities of potential side effects that might affect the immune response, we have experimentally excluded the contribution of Ly6C^+^ cells in Figure 9 and Figure 9—figure supplement 1. We have also added a statement to emphasize this fact in the subsection “LN-resident, but not migratory, DCs promote Tfh, GC B and antibody responses”.

5) The authors have not addressed concerns regarding the contribution of PDL1 in CD103b+ DC-mediated control of Tfh and B cell response. Please remove strong claims regarding the potential role of PDL-1 from the Abstract and modulate their claim in this regard in the Results and Discussion sections.

We modified the Abstract and deleted "possibly through PD-L1" from the Results. We also modified our statement in the Discussion (sixth paragraph) and also mentioned the possibility of PD-1-independent mechanism (seventh paragraph).